# Feasibility, safety and tolerability of estrogen and/or probiotics for improving vaginal health in Canadian African, Caribbean, and Black women: A pilot phase 1 clinical trial

Biban Gill[1], Jocelyn M. Wessels[1,2], Christina L. Hayes[1], Jenna Ratcliffe[1], Junic Wokuri[3], Elizabeth Ball[1], Gregor Reid[4], Rupert Kaul[5,6], Jesleen Rana[3], Muna Alkhaifi[3], Wangari Tharao[3], Fiona Smaill[7] *, Charu Kaushic[1] *

1 McMaster Immunology Research Centre and Department of Medicine, McMaster University, Hamilton, ON, Canada, 2 Afynia Laboratories, Hamilton, ON, Canada, 3 Women's Health in Women's Hands Community Health Centre, Toronto, ON, Canada, 4 Departments of Microbiology & Immunology and Surgery, Western University, and Canadian Research and Development Centre for Human Microbiome and Probiotics, The Lawson Health Research Institute, London, ON, Canada, 5 Departments of Immunology and Medicine, University of Toronto, Toronto, ON, Canada, 6 Department of Medicine, University Health Network, Toronto, ON, Canada, 7 Department of Pathology and Molecular Medicine and Michael G. DeGroote Institute for Infectious Disease Research, McMaster University, Hamilton, ON, Canada

* kaushic@mcmaster.ca (CK); smaill@mcmaster.ca (FS)

## Abstract

### Background

A dysbiotic vaginal microbiome (VMB) is associated with clinical conditions such as bacterial vaginosis (BV) and an increased risk of human immunodeficiency virus (HIV-1) infection. Considering the high prevalence of BV among African, Caribbean and Black (ACB) women, we conducted a prospective, randomized, open-label phase 1 clinical trial to determine the feasibility, safety and tolerability of administering low-dose estrogen, probiotics or both in combination to improve vaginal health and decrease HIV-1 susceptibility.

### Methods

ACB women aged 18–49 from the Greater Toronto Area (GTA) were randomized to one of four study arms: intravaginal estradiol (Estring©; 7.5mg/day); a vaginal probiotic (RepHresh™ Pro-B™) administered twice daily; a combination of Estring© and vaginal RepHresh™ Pro-B™ (twice daily); or the Estring© and oral RepHresh™ Pro-B™ (twice daily), for a duration of 30 days. Feasibility was evaluated through enrolment, retention, and adherence rates, while safety and tolerability were determined by a pre- and post-treatment blood panel and reported adverse events (AEs).

### Results

Overall, 63 ACB women were screened, 50 were enrolled and received the intervention while 41 completed the study, resulting in 80% enrollment and 82% retention rates. Overall adherence to the study protocol was high at 93%, with an adherence of 92% for RepHresh™

**Funding:** This study was funded by CIHR Grants FRN#159229 and FRN#154047 (CK) and in-kind contributions from CIHR HIV Clinical Trials Network (FS, CK). B.G was partially supported by a Post-doctoral Award from CIHR FRN#181875. There was no additional external funding received for this study. The funders had no role in study design, data collection and analysis, decision to publish, or preparation of the manuscript.

Pro-B™ and 97% for Estring©. A total of 88 AEs were reported by 29 participants which were mild (66/88; 75%) and largely resolved (82/88;93%) by the end of the study, with no serious AEs (SAEs) noted. In addition, a panel of safety blood markers measured pre- and post-intervention confirmed no clinically significant changes in blood chemistry or blood cell count.

## Conclusion

Overall, the administration of intravaginal estrogen and/or probiotics in pre-menopausal ACB women is feasible, safe, and well tolerated.

## Trial registration

The trial was registered with Clinicaltrials.gov (NCT03837015) and CIHR HIV Clinical Trials (CTN308).

## Introduction

Despite advancements in treatment and care, Human Immunodeficiency Virus-1 **(HIV-1)** remains a significant global health concern, with women bearing a disproportionate burden [1]. African, Caribbean, and Black **(ACB)** women in Canada experience new infection rates seven times higher than that of Caucasian women [2]. Considering that 33% of new HIV-1 infections among ACB Canadians are attributed to heterosexual transmission [3] it is critical we investigate modifiable factors which can decrease HIV-1 risk and inform prevention. Specifically, the composition of vaginal microbiota within the lower female genital tract **(FGT)** is considered a central mediator of biological susceptibility [4, 5]. For instance, a dysbiotic vaginal microbiome **(VMB),** characterized by increased microbial diversity and a reduction in *Lactobacillus spp*. has been associated with vaginal inflammation, an increased incidence of sexually transmitted infections **(STI)** and clinical conditions such as bacterial vaginosis **(BV)** [4, 6–8]. Moreover, the presence of BV, which is more prevalent in women from ACB communities [9–11], has been consistently associated with a 2-5-fold increase in HIV-1 risk [12, 13]. As a result, there is a push for research in this area to better understand how to prevent and treat BV in order to reduce biological susceptibility to HIV-1 among at-risk populations [9, 14].

Although antibiotics are currently considered the standard of care for BV [9, 15], there are high rates of recurrence post-treatment [9, 16–19]. Consequently, *Lactobacillus*-based probiotics have been extensively explored as a complementary or sole therapeutic to restore and maintain a healthy VMB [16, 20–26]. In particular, oral probiotics when combined with antibiotics have been shown to facilitate the displacement of BV-related anaerobes, improve cure rates and relieve the symptoms of BV [24, 27, 28]. However, intravaginal administration of probiotic strains have the advantage of rapid and direct access to the local environment as well as enhancing the local immune defenses in the lower FGT [29, 30]. A recent study of vaginally administered biotherapeutic *Lactobacillus crispatus* CTV-05 strain LACTIN-V, demonstrated reduced BV recurrence at 24 weeks from 54% to 39% [31]. While the use of a *Lactobacillus* species that are commonly found in the healthy vagina makes logical sense, not all *L. crispatus* strains are suitable to promote vaginal health [32]. Therefore, an important consideration for an intervention to improve vaginal health should be the bacterial strain's ability to inhibit and displace urogenital pathogens [33, 34]. Of the strains documented for possessing these

properties, Lacticaseibacillus (formerly Lactobacillus) rhamnosus GR-1 and Limosilactobacillus (formerly Lactobacillus) reuteri RC-14 are the most frequently selected across trials and are most often consumed as an oral supplement [21, 30, 31, 33]. The ability of L. rhamnosus GR-1 to enhance the vaginal epithelial barrier [29], produce acid that kills similar viruses [33] and its availability in a commercial product (RepHresh™ Pro-B™) makes this a good choice for testing. Despite providing therapeutic benefits and a reduction in BV episodes, the effects are often not long-lasting [26, 34] with few studies having explored these strains following vaginal administration beyond five days [35, 36]. This highlights the need to either use the probiotic for a longer period or develop other strategies to establish and maintain a woman's protective *Lactobacillus* species.

In the context of HIV-1 risk, female sex hormones in the lower FGT have also been extensively studied [4, 37–39]. Estrogen has been shown to decrease inflammation, increase epithelial barrier integrity and promote *Lactobacillus* growth factors that enhance protection against infection in *in vitro* studies [4, 40]. Estrogen increases the production of glycogen in vaginal epithelial cells, and bacterial amylases enable glycogen breakdown, promoting *Lactobacillus* persistence [4, 41]. This concept has been widely applied among post-menopausal women for the treatment of urogenital disorders where low-dose estrogen-containing vaginal rings (**Estring**©) are able to restore estrogen levels, increase vaginal pH and enhance *Lactobacillus* persistence [36, 42–44]. While the Estring© has not been used in premenopausal women, both topical estrogen creams and vaginal rings containing estrogen have been employed safely in post-menopausal women and are well-tolerated [5, 43]. By delivering local estrogen in conjunction with a probiotic we are testing an innovative strategy to enhance and sustain the persistence of *Lactobacillus* strains in the vaginal microbiota, with the aim of reducing HIV-1 risk among high-risk populations.

The study objective was to perform a prospective, randomized, open-label, intervention study among healthy, premenopausal ACB women to assess the feasibility, safety, and tolerability of administering low-dose intravaginal estrogen (Estring©) and/or probiotics (RepHresh™ Pro-B™). Primary outcome measures including enrollment, retention and adherence were collected to evaluate feasibility, while a panel of blood markers and data on adverse events (**AEs**) were monitored to establish safety and tolerability.

## Materials and methods

The protocol and supporting CONSORT checklist are available as supporting information; see **S1 Checklist** and **S1 Protocol**.

### Ethics approval

This study was conducted according to the International Conference on Harmonisation Good Clinical Practices (ICH GCP) guidelines, applicable Health Canada regulations and the principles of the Declaration of Helsinki. Ethics approval was obtained from the Hamilton Integrated Research Ethics Board (HiREB Project #7061), and the trial was registered with Clinicaltrials. gov (NCT03837015) and CIHR Canadian HIV Clinical Trials Network (CTN 308). Recruitment for the study happened between Nov 1, 2019 and Dec 31, 2021. Participants received a detailed oral and written description of the study's interventions, procedures, and risks which was reviewed with them by the Research Study Nurse. They were given an opportunity to ask questions and consider if they wanted to participate in the study before they provided written informed consent at the time of recruitment. The informed consents were reviewed during onsite monitoring visits by CIHR HIV CTN Staff. Participants were compensated for each study visit for the costs of travel, parking, child-care and loss of work time. Study oversight

and support, data oversight and study monitoring were provided by the CIHR Canadian HIV Trials Network (CTN) and their Data Safety Monitoring Board (DSMB).

## Study design

The present study was designed as a 30-day, prospective, randomized, open-label, phase 1 trial to evaluate the feasibility, safety and tolerability of administering low-dose intravaginal estrogen and/or probiotics. Healthy, premenopausal ACB women between 18–49 years of age were recruited from the Greater Toronto Area (GTA) between November 2019 and December 2021 with clinic visits carried out at the Women's Health in Women's Hands (WHIWH) Community Centre in Ontario, Canada. Individuals were enrolled according to the inclusion and exclusion criteria outlined in Table 1 (Full details in Study S1 Protocol) and those who were deemed eligible were assigned to one of four study arms using block randomization with a 1:1:1:1 ratio and block sizes of either four or eight (Fig 1). This process was facilitated through an electronic code generator and securely stored by Bay Area Research Logistics (BARL),

**Table 1. Summary of study eligibility criteria.**

| Inclusion Criteria | Exclusion Criteria |
|---|---|
| ✓ Women 18–49 years of age, inclusive | ✕ Currently lactating |
| ✓ African, Caribbean, Black | ✕ Pregnant: suspected, current or in the last 12 months |
| ✓ Pre-menopausal women in good general health, | ✕ Irregular menstrual cycle |
| ✓ Uterus and cervix present | ✕ Post-menopausal |
| ✓ Negative pregnancy test | ✕ Hormonal contraceptive use or other hormonal treatment in the past 3 months |
| ✓ Currently practicing barrier or non-hormonal forms of contraception, and planning to continue | ✕ Current Intra-Uterine Device (IUD) use |
| ✓ Willing to undergo a pelvic exam by a female nurse | ✕ Positive test result for Gonorrhea and/or Chlamydia |
| ✓ Willing to abstain from vaginal intercourse for 48 hours prior to sampling, over the entire course of the study | ✕ Clinically obvious genital ulceration/lesions |
| ✓ Able to understand, comply and consent to protocol requirements and instructions | ✕ Symptomatic vaginal yeast infection |
| ✓ Able to attend scheduled study visits and complete required investigations | ✕ HIV-positive |
| | ✕ Fail screening safety blood tests |
| | ✕ Diagnosed blood clotting disorder |
| | ✕ Any genital tract procedure within the past 6 months |
| | ✕ Use of oral probiotic supplement, oral antibiotics or oral steroids within the past 30 days |
| | ✕ Current use of any vaginal products (except tampons) |
| | ✕ Known intolerance of Lactobacillus-containing probiotics |
| | ✕ Undiagnosed abnormal genital bleeding |
| | ✕ Known, suspected, or history of breast cancer |
| | ✕ Known or suspected estrogen-dependent malignant neoplasia |
| | ✕ Currently taking immunosuppressive drugs |
| | ✕ Known or suspected hypersensitivity to any component of the Estring or RepHresh Pro-B products |
| | ✕ Diagnosis of endometrial hyperplasia |
| | ✕ Known liver dysfunction or disease |
| | ✕ Active or past history of arterial thromboembolic disease |
| | ✕ Partial or complete vision loss |
| | ✕ Porphyria |
| | ✕ Concomitant medication and medical conditions |

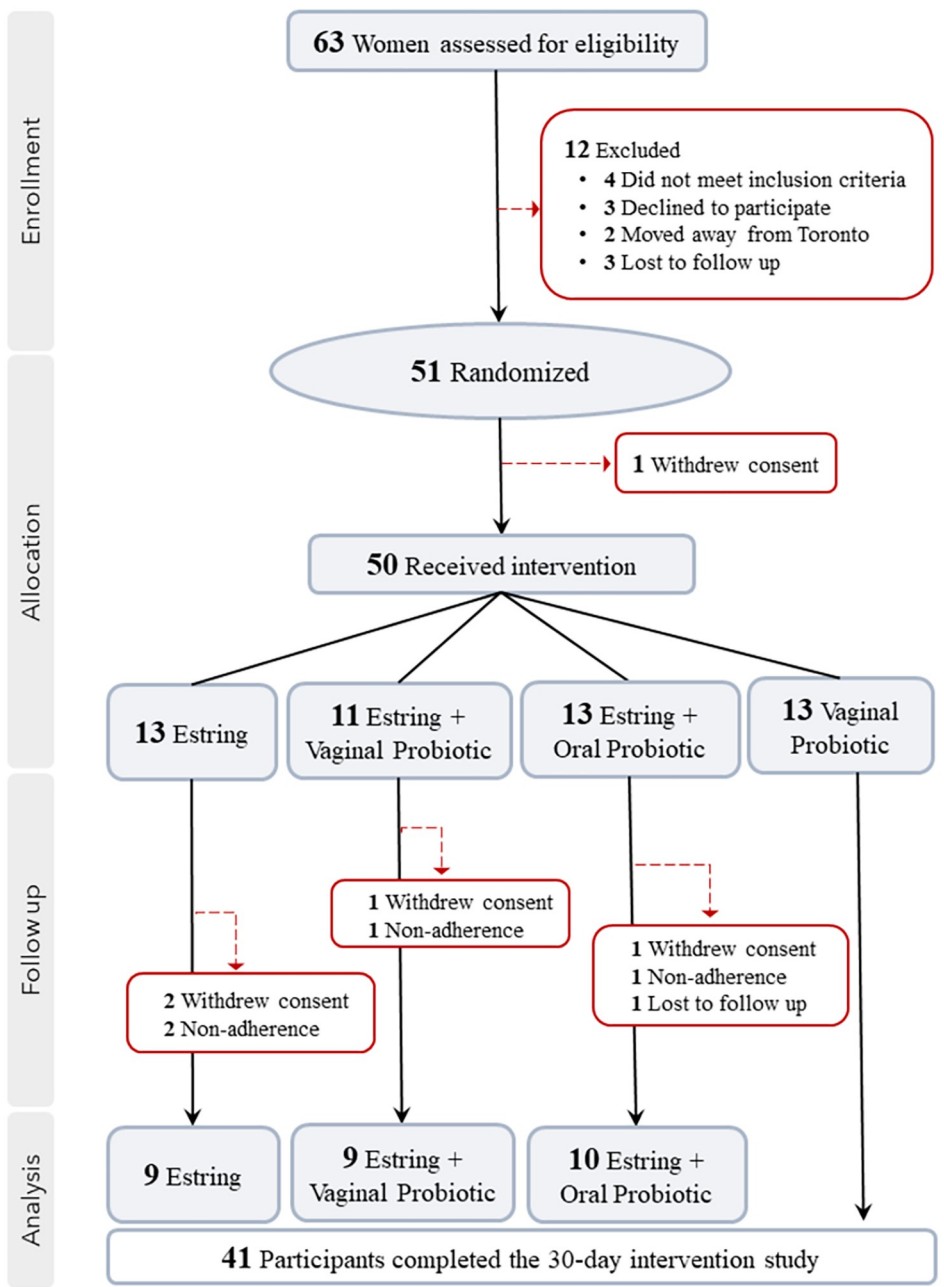

**Fig 1. Participant flowchart.** A flow diagram outlining study recruitment and completion rates. A total of 63 women were assessed for eligibility and 50 women received intervention. One woman was lost to follow up, four withdrew consent and an additional four were non-adherent to the study protocol. Overall, 41 women completed the study with group sizes between 9–13 participants for each study arm.

Hamilton. The study kits were also prepared by BARL and labelled numerically according to the assigned groupings (Groups 1–4). The nurse was responsible for dispensing the kits and only upon opening were the participant and nurse made aware of the designated intervention. Other members of the study team remained blinded until all the results had been analyzed.

The study design is presented in **Fig 2** and the information collected at each visit is summarized in **Table 2**. Briefly, a total of four in clinic visits, with an optional fifth follow-up visit

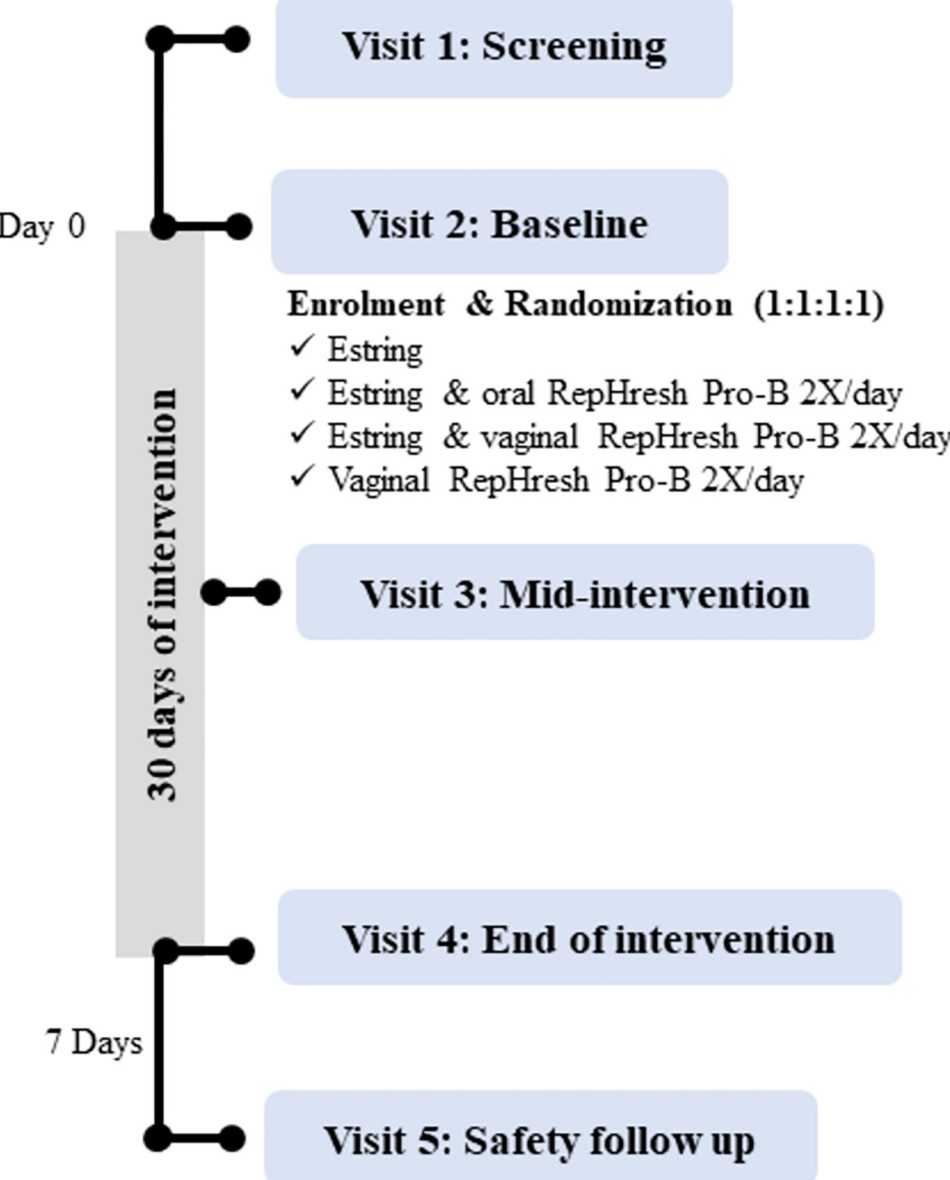

**Fig 2. Study design.** This schematic outlines the design and study timeline used to evaluate the feasibility, safety and tolerability of administering low dose estrogen and/or a probiotic to premenopausal African, Caribbean, and Black (ACB) women. Eligible participants were randomized in a 1:1:1:1 ratio to receive one of four interventions including 1) the Estring alone 2) the Estring in combination with the probiotic *L. rhamnosus* GR-1 and *L. reuteri* RC-14 RepHresh Pro-B (administered vaginally), 3) the Estring together with the RepHresh Pro-B (administered orally) or the RepHresh (administered vaginally) alone. The duration of the intervention was 30 days with a 7-day follow up (visit 5) safety check. The study was designed to compare outcomes in each woman at baseline (visit 2) to the end of treatment (visit 4) with each participant serving as their own control.

were conducted. During visit 1 (screening) informed consent, eligibility, medical history, a physical exam, pregnancy test, and STI screening were completed. In addition, blood was collected to evaluate a panel of safety markers pre-intervention. Eligible participants were then randomized and enrolled at visit 2 (Baseline; Day 0) during which time the study products were provided, physical examinations, questionnaires (Sexual History Questionnaire), diary review, assessment of adverse events **(AE)**, pregnancy tests and biological sample collection

**Table 2. Schedule of events outlining study procedures and data collection by visit.**

| | Visit 1 | Visit 2 | Visit 3 | Visit 4 | Visit 5 |
|---|---|---|---|---|---|
| | Screening | Baseline[a] | Mid-Intervention | End of Intervention | Safety follow up[b] |
| **Visit day** | - | 0 | 14± 3 | 31± 3 | 37± 3 |
| **Informed consent** | ✓ | | | | |
| **Assessment of eligibility** | ✓ | ✓ | | | |
| **Randomization** | | ✓ | | | |
| **Clinical evaluation** | | | | | |
| Medical history | ✓ | | | | |
| Concomitant medication | | ✓ | ✓ | ✓ | ✓ |
| Physical exam[c] | ✓ | ✓ | ✓ | ✓ | ✓ |
| Pelvic exam | ✓ | ✓ | ✓ | ✓ | ✓ |
| **Clinical laboratory tests** | | | | | |
| STI testing[d] | ✓ | | | | |
| Urine pregnancy test | ✓ | ✓ | | | |
| Safety blood (20mL) | ✓ | | | ✓ | |
| HIV serology | ✓ | | | | |
| **Sexual history questionnaire** | | ✓ | | | |
| **Study product dispensation** | | ✓ | | | |
| **Blood collection (20–40 mL)[f]** | | ✓ | | ✓ | |
| **Nurse-collected vaginal swabs** | | | | | |
| PSA testing[g] | | ✓ | ✓ | ✓ | ✓ |
| BV testing | | ✓ | | | |
| Microbiome/HSV-2 | | ✓ | ✓ | ✓ | ✓ |
| **Self-collected vaginal swabs[e]** | | ✓ | ✓ | ✓ | |
| **Cervical sampling** | | | | | |
| CVLs | | ✓ | ✓ | ✓ | ✓ |
| Cytobrush | | ✓ | ✓ | ✓ | ✓ |
| **Assessment of AEs** | | ✓ | ✓ | ✓ | ✓ |
| **Study diaries** | | | | | |
| Distribution | | ✓ | | | |
| Review | | | ✓ | ✓ | ✓ |
| Return | | | | ✓ | ✓ |
| **Return of study products** | | | | ✓ | |
| **Assessment of compliance** | | | ✓ | ✓ | |

a. Scheduled 5–10 days after 1st day of menstrual cycle

b. See protocol for procedure to be completed if participant has consented to attending an "in-clinic" visit.

c. A directed physical examination will be performed at subsequent visits after the screening visit

d. Urine NAAT

e. Self-collected vaginal swab will be done during visit 2, and a maximum of 24 hrs prior to visits 3 and 4

f. 20–40 mL blood (four tubes) collected for PBMCs for immunology phenotyping, DNA isolated for genetic analysis, plasma for chemokine/cytokine/immune protein/hormone detection, HSV-2 screening at baseline only

g. Prostate-specific antigen (PSA) test to determine unprotected sex in past 48 hours. Swab collected prior to collection of cervical samples.

were completed. Blood samples were then tested for safety markers. At each subsequent study visit, physical examinations were repeated, and participants provided updates on AEs, health changes, new medical conditions, and any medication changes. Biological samples collected at multiple time points over the course of the study included blood, urine, cervicovaginal lavages (**CVLs**) as well as both nurse and self-collected vaginal swabs (**Table 2**). Samples were used for

clinical and research laboratory tests as described in **S1 Protocol**. Baseline BV status (Nugent score), and recent unprotected intercourse using a rapid prostate-specific antigen **(PSA)** testing kit (Seratec PSA Semiquant, Gottingen, Germany) were assessed. Midway through the intervention participants completed visit 3 (Mid-Intervention) in the clinic as described above with an additional assessment for protocol adherence. Subsequently, at visit 4 (End of Intervention) participants underwent an additional blood collection for evaluation of safety markers post-intervention, alongside the routine assessment described previously (**Table 2**). At this time diaries were returned, and all unused study products, as well the product containers were returned to the nurse. Lastly, a 7-day safety follow-up (visit 5) was completed by phone or in clinic if the participant consented to an additional biological sample collection.

## Study products

Low-dose estradiol (Estring©) was administered vaginally alone or in combination with a *L. rhamnosus* GR-1 and *L. reuteri* RC-14 probiotic (RepHresh™ Pro-B™) that was delivered twice daily either orally or vaginally. The Estring© released 2mg of estradiol at a rate of 7.5ug/24 hours and had an outer diameter of 55 mm and cross-section diameter of 9.5mm [41]. The RepHresh™ Pro-B™is a patented probiotic feminine supplement containing $1\times10^9$ cfu total of *L. rhamnosus* GR-1 and *L. reuteri* RC-14 per capsule [27]. Interventions were distributed across the four study arms as follows: Groups 1–3 were given a single packet containing one Estring© vaginal ring and instructed to keep it in place for the 30-day intervention. Groups 2 and 3 were told to also take the RepHresh™ Pro-B™ capsules twice daily, roughly 12 hours apart over 30 days. Importantly, the route of probiotic administration differed between Groups 2 and 3, where Group 2 administered the probiotic intravaginally, and Group 3 consumed the capsules orally. Lastly, Group 4 was exclusively told to administer the probiotic vaginally as described above.

## Outcome measures

The primary outcome measures were feasibility, safety, and tolerability. Secondary and exploratory outcomes are outlined in **S1 Protocol** and will be reported in subsequent studies. Feasibility was assessed according to the proportion of eligible participants who consented (enrollment rate) and completed the study (retention rate), along with the intervention protocol adherence, and completion of diaries and questionnaires (Sexual History Questionnaire). Adherence was determined according to either the percentage of probiotics consumed out of the total number dispensed, or the percentage of days the Estring© was used relative to the study's duration. Adherence was evaluated among participants who received the intervention (n = 50), including individuals who withdrew consent, were lost to follow up or were non-adherent to the intervention protocol and did not complete the study. When including participants who did not complete the intervention, they were assigned a value of zero. Safety and tolerability were evaluated based on a panel of blood markers which included blood glucose, complete blood count as well as comprehensive metabolic and lipid panels taken at baseline (visit 2) and the end of the intervention (visit 4). In addition, AEs were investigated according to reported intensity (mild, moderate, severe) [45], relationship to the type of intervention (probably related, possibly related, unlikely to be related, or not related) [46], frequency and the proportion of participants experiencing such events. AEs were defined as any untoward medical occurrence in a participant which does not necessarily have a causal relationship with the intervention (**S1 Protocol**) [45]. These events were distinguished from serious adverse events **(SAEs)** which are described as an AE that results in any of the following; death during the period of the protocol, a life-threatening event, significant disability or incapacity, a congenital anomaly, or in-patient hospitalization (**S1 Protocol**) [45].

## Statistical analysis

Statistical analysis was completed using GraphPad version 6.04, and R version 4.2.3. Significance tests were two-sided, and significance was taken at $p < 0.05$. Normality was determined according to the Shapiro–Wilk test. Descriptive statistics are presented as either mean (SD) or median (IQR) for continuous variables and as numbers and percentages for categorical variables. Differences between study visits were determined with a paired t-test or Wilcoxon signed-rank test if appropriate, while comparisons between the interventions were investigated using an ANOVA or a Kruskal-Wallis test. All analyses were carried out based on an intention-to-treat **(ITT)** principle with no missing data imputed.

## Results

### Participant demographics

Sixty-three women were screened for eligibility and 51 were randomized across the four study arms (**Fig 1**)**.** One participant withdrew consent prior to the baseline visit resulting in 50 women who received an intervention thereby representing the ITT population utilized for subsequent analysis (**Fig 1**). Baseline characteristics and demographics across interventions are summarized in **Table 3**. The majority (35/50; 70%) of participants were born outside of Canada and immigrated during adulthood; the average age across study groups was 35 (28–39 years). Most women (47/50; 94%) reported a regular menstrual cycle, and few (7/50; 14%) reported gynecological symptoms within the past month. Over half (27/50; 54%) of participants reported having had vaginal sex in the past six months with a slight minority (12/27; 44%) using a condom >50% of the time. Out of the 50 women enrolled 24% (12/50) reported a history of BV based on clinical symptoms, half (6/12; 50%) of whom experienced multiple (>1) episodes. Overall, the distribution of BV (determined by Nugent score) at baseline was not equally distributed across study arms with the oral probiotic group contributing to the majority of cases (8/10). Notably, the women that noted experiencing a persistent recurrence of clinical BV also exhibited evidence of BV by Nugent score at the initial in-clinic evaluation, contributing to 60% (6/10) of the BV diagnoses observed at baseline (visit 2).

### Feasibility outcomes

Eighty percent of screened participants were enrolled and received the intervention (50/63) while 82% (41/50) were retained at the end of the intervention (visit 4), exceeding or meeting targets of 70% and 80%, respectively (**Table 4**). Notably, the highest rate of retention was observed among those administering the vaginal probiotic where 100% completed the study. The nine participants who did not complete the study all terminated between baseline (visit 2) and mid-intervention (visit 3). Four women withdrew consent (4/50; 8%), four self-reported non-adherence (4/50; 8%) and one was lost to follow-up (1/50; 2%; **Table 4**).

Participants who received the intervention demonstrated a consistent and high adherence irrespective of intervention type or route of administration (i.e. oral vs. vaginal probiotic) with an overall adherence of 93% (IQR;89,99), a probiotic adherence of 92% (IQR; 87,100), and an Estring© adherence of 97% (IQR; 52,100) as shown in **Fig 3**. Although no significant differences were observed between study arms the Estring-only participants demonstrated a lower adherence (83%; IQR 0,100%) relative to the combined interventions (97–100%; **Table 4**) as a result of four participants terminating the study prematurely (**Fig 3B**). For probiotic users, adherence was similar between study arms with the group who received the combined Estring© and oral probiotic exhibiting a slightly lower adherence (88%;IQR 77,97) as

**Table 3. Baseline characteristics and demographics for participants who received intervention.**

| Characteristic | ER (n = 13) | ER+vP (n = 11) | ER+oP (n = 13) | vP (n = 13) | Overall (n = 50) |
|---|---|---|---|---|---|
| Age, years, mean (SD) | 33.7 (7.3) | 34.2 (8.5) | 33.5 (6.9) | 36.8 (6.4) | 34.5 (7.2) |
| BMI, kg/m$^2$, mean (SD) | 27.6 (5.2) | 29.5 (5.9) | 29.2 (9.2) | 28.5 (7.8) | 28.7 (7.0) |
| ACB Ethnicity, n (%) | 13 (100) | 11 (100) | 13 (100) | 13 (100) | 50 (100) |
| Region of birth, n (%) | | | | | |
| North America | 1 (8) | 6 (55) | 6 (46) | 3 (23) | 16 (32) |
| Caribbean | 4 (31) | 1 (9) | 2 (15) | 2 (15) | 9 (18) |
| Africa | 8 (62) | 4 (36) | 5 (39) | 8 (62) | 25 (50) |
| Relationship status, n (%) | | | | | |
| Single | 8 (62) | 6 (55) | 6 (46) | 5 (39) | 25 (50) |
| Married | 1 (8) | 4 (36) | 2 (15) | 4 (31) | 11 (22) |
| Other | 4 (31) | 1 (9) | 5 (39) | 4 (31) | 14 (28) |
| Employment status, n (%) | | | | | |
| Employed full-time | 2 (15) | 1 (9) | 4 (31) | 4 (31) | 11 (22) |
| Employed part-time | 2 (15) | 4 (36) | 3 (23) | 4 (31) | 13 (26) |
| Unemployed | 3 (23) | 3 (27) | 3 (23) | 3 (23) | 12 (24) |
| Other | 5 (38) | 3 (27) | 3 (23) | 2 (15) | 13 (26) |
| Education, n (%) | | | | | |
| Completed college or university | 6 (46) | 6 (60) | 6 (46) | 8 (62) | 26 (53) |
| Completed secondary/high school | 1 (8) | 2 (20) | 1 (8) | 0 (0.0) | 4 (8) |
| Completed elementary/primary | 0 (0) | 1 (9) | 0 (0) | 0 (0) | 1 (2) |
| Regular period, n (%) | 13 (100) | 10 (91) | 13 (100) | 11 (85) | 47 (94) |
| Pregnant in the past, n (%) | 6 (46) | 7 (64) | 9 (69) | 9 (69) | 31 (62) |
| Sexual partners in the past 6 months, n (%) | | | | | |
| 0 | 7 (54) | 5 (46) | 5 (38) | 5 (38) | 22 (44) |
| 1–5 | 5 (38) | 6 (55) | 8 (62) | 8 (62) | 27 (54) |
| Chose not to respond | 1 (8) | 0 (0) | 0 (0) | 0 (0) | 1 (2) |
| Past history of STIs, n (%) | 2 (16) | 4 (36) | 5 (39) | 1 (8) | 12 (24) |
| Reported BV diagnosis history, n (%) | | | | | |
| 1 diagnosis | 3 (23) | 2 (18) | 0 (0) | 1 (8) | 6 (12) |
| >1 diagnosis | 1 (8) | 1 (9) | 1 (8) | 3 (23) | 6 (12) |
| BV at baseline (Visit 2) | 0 (0) | 1 (10) | 4 (40) | 4 (31) | 10 (21) |

Data shown as n (% of participants) or mean±SD, unless stated otherwise. ANOVA and/or a Kruskal wallis test was to evaluate significant differences between treatment arms.

ER: Estring; ER+vP: Estring and vaginal probiotic; ER+oP Eststring and oral probiotic; vP: vaginal probiotic

ACB: African/Carribean/Black

compared to the two additional study groups receiving the vaginal probiotic interventions (92–93%).

To determine study feasibility, additional adherence parameters were evaluated according to important factors within the study procedures. This included the completion of the sexual history questionnaires and diaries (Table 4). While 100% of participants filled out the sexual history questionnaire, a small reduction in the completion of specific questionnaire items such as sexual orientation, number of sexual partners within a lifetime and during the last six months was observed. Completion of questionnaire items remained consistent between study groups. Similarly, diary completion across all visits and interventions was 100% among those

**Table 4. Summary of study participation and feasibility.**

| | ER (n = 13) | ER+vP (n = 11) | ER+oP (n = 13) | vP (n = 13) | Overall |
|---|---|---|---|---|---|
| **Study participation, n (%)** | | | | | |
| Visit 1 (Screening) | 13 (100) | 11 (100) | 13 (100) | 13 (100) | 63 (100) |
| Visit 2 (Baseline) | 13 (100) | 11 (100) | 13 (100) | 13 (100) | 50 (80) |
| Visit 3 (Mid-Intervention) | 10 (77) | 8 (72) | 10 (76) | 13 (100) | 41 (82) |
| Visit 4 (End of Intervention) | 9 (69) | 9 (81) | 10 (76) | 13 (100) | 41 (82) |
| Visit 5 (Follow up) | 9 (69) | 9 (81) | 10 (76) | 13 (100) | 41 (82) |
| **Participant termination, n (%)** | | | | | |
| Lost to follow up | 0 (0) | 0 (0) | 1 (7) | 0 (0) | 1 (2) |
| Withdrew consent | 2 (15) | 1 (9) | 1 (7) | 0 (0) | 4 (8) |
| Non-adherence | 2 (15) | 1 (9) | 1 (7) | 0 (0) | 4 (8) |
| **Adherence (%), [a] median (IQR)** | | | | | |
| Estring | 83 (0, 100) | 97 (91, 100) | 100 (75, 100) | - | 97 (52, 100) |
| Probiotic | - | 92 (32, 100) | 88 (77, 97) | 93 (90, 98) | 92 (87, 100) |
| Diary | 97 (0, 100) | 97 (94, 1.00) | 97 (94, 98) | 100 (97, 100) | 97 (94, 100) |
| Questionnaire completion, n (%) | 12 (92) | 11 (100) | 13 (100) | 13 (100) | 49 (98) |
| PSA, mean ± SD | 92 ± 15 | 93 ± 16 | 91 ± 16 | 77 ± 24 | 88 ± 19 |

[a]Participant who did not complete the protocol was considered as nonadherent and a value of zero was assigned for the various adherence measures.

Enrollment rate (# enrolled ÷ # screened)

Retention rate (# completed ÷ # enrolled)

Data shown is presented as n (% of participants) or median (IQR) unless stated otherwise.

The n value for the total enrolled for each intervention type is indicated in brackets (n =)

Adherence = # of days/doses of intervention participant used ÷ total # of days/doses in protocol

(30 days of Estring; 60 probiotic doses)

PSA adherence is based on positive PSA tests across all visits

PSA = prostate-specific antigen

ER: Estring; ER+vP: Estring and vaginal probiotic; ER+oP Eststring and oral probiotic; vP: vaginal probiotic

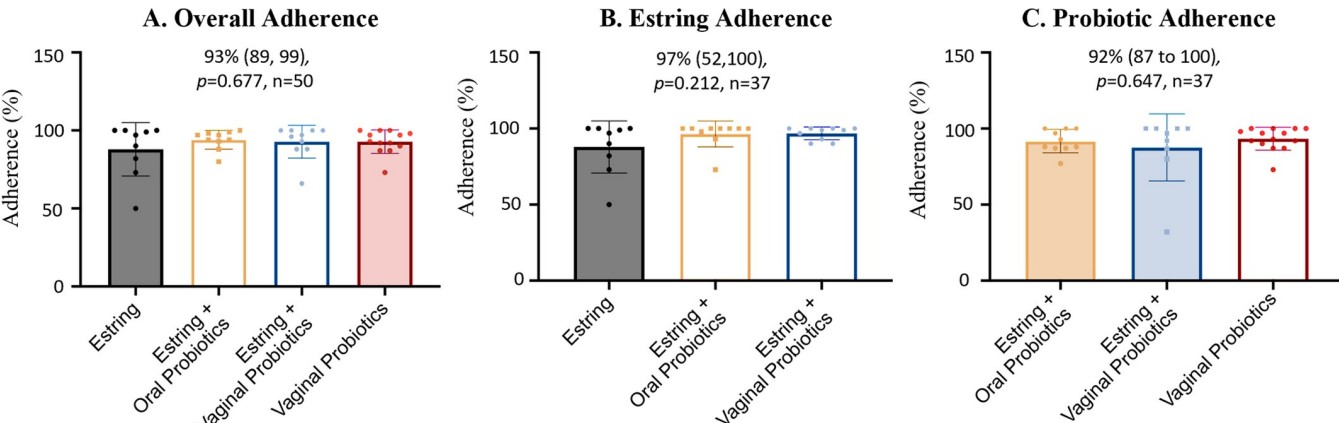

**Fig 3. Intervention protocol adherence.** Adherence was determined for Estring and probiotic intervention protocols (IP) both individually and overall. The bar plots indicate adherence for all participants who received the intervention including those who prematurely terminated the study. Women who did not complete the protocol were considered as nonadherent and assigned a value of zero. Each data point represents a participant, and the bars depict the median ± IQR. **A.** Overall participants showed a high adherence to the study protocols regardless of the study. **B-C.** Similarly, participants demonstrated high adherence for both Estring and probiotic administration with no significant differences observed between intervention type or route of probiotic administration.

who completed the study and 97% when including participants who discontinued the intervention (**Table 4**). The completion rates remained consistent (97–100%) across study groups and between visits (98–100%). As part of the study protocol, women were asked to abstain from unprotected vaginal intercourse 48 hours before in clinic study visits. However, prostate specific antigen(PSA) was detectable (positive test result) in 13% (21/156) of all vaginal swabs across visits, suggestive of recent unprotected vaginal intercourse, attributed to 16 (16/50; 32%) participants with a minimum of one PSA positive result. Overall, 48-hour abstinence adherence was 88% ±19, with the vaginal probiotic group exhibiting the lowest at 77%± 24 relative to an average at 91–93% for the remaining study arms (**Table 4**). Importantly, no significant association between PSA and study visit or intervention type was detected, and all PSA-positive participants completed the study. A detailed summary of adherence measures across study groups is outlined in **Table 4**.

## Safety and tolerability outcomes

A total of 88 AEs were reported by 37 participants who used each intervention across the four study groups (**S1 Table in S1 File**). Overall, no serious AEs (SAEs) were reported, as per NIH Department of AIDS AE guidelines used in this study [45], and no indications of inflammation or infection were observed during the pelvic exams. Most AEs were mild in intensity (66/88; 75%), short-term (82/88; 93% resolved by end of study), and non-recurring (**S1 Table in S1 File**). Vaginal irritation/burning/itching (20/88;22%), abdominal pain/cramps (12/88;14%), and headache (11/88; 12%) were the most reported AEs (**Fig 4**). A total of three participants reported AEs of severe intensity including cramps/abdominal pain, headache, lightheadedness and/or nausea which were all resolved by study completion (**S1 Table in S1 File**). Insomnia, vaginal irritation/itching, breast tenderness, and headache were the only ongoing AEs (6/88; 7%) at the end of the study for three participants (**S1 Table in S1 File**). Twenty-nine (29/88; 33%) AEs were associated with an interruption to the treatment protocol with one participant

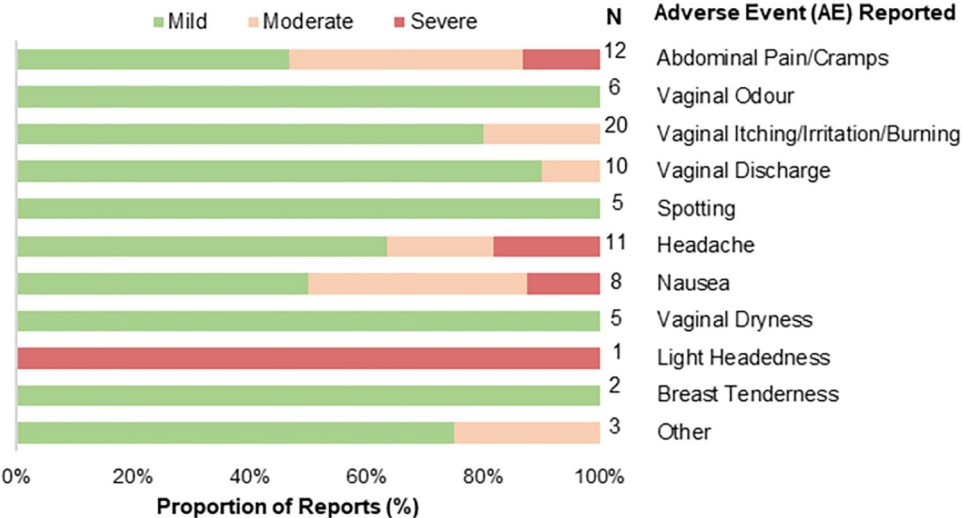

**Fig 4. Adverse events (AEs).** A total of 88 AEs were reported by 37 participants for each intervention. This bar plot summarizes the types of AEs based on their reported frequency and intensity. AEs reported with a mild intensity are represented in green while a moderate intensity is indicated in orange and severe AEs are highlighted in red. The number of reports according to event type is outlined based on the frequency of reports. Most reported AEs were mild (66/88; 75%), with only 7% (6/88) showing a severe intensity and no serious AEs (SAEs) reported. Vaginal irritation/ itching/burning followed by cramps/abdominal pain were the most frequently reported AEs. In additional most AEs were short lived and non-recurring with 7% (6/88) ongoing after study completion.

from the Estring©+ vaginal probiotic group reporting four (4/88; 4%) AEs and choosing to discontinue the intervention (**S2 Table in S1 File**).

No clinically relevant changes in vital signs including pulse, blood pressure, and temperature were observed at baseline relative to the end of treatment. Safety blood markers for general health and immune cells across study arms before and after intervention are summarized in **Fig 5**. The data obtained before and after the 30 days of intervention demonstrated no clinically significant changes in metabolite, lipid and complete blood count panels (**Fig 5**). Moreover, a consistent distribution and range in the datasets was exhibited across the study arms for each blood marker. The normal and healthy clinical ranges for each blood marker are indicated in **Fig 5**. Importantly, participants with values outside of these thresholds were not associated with an intervention type and were randomly distributed. Overall, the markers reflect an acceptable biological variation across fifty participants with certain individuals falling outside of recommended clinical thresholds which is to be expected considering variation in age, health, lifestyle, diet, medication use and comorbidities among the population.

## Discussion

This trial successfully demonstrated the safety and feasibility of administering low-dose intravaginal estrogen and/or probiotics to Canadian ACB premenopausal women. Notably, this is the first study to explore the extended use (>5 days) of the Estring© and vaginally administered RepHresh™ Pro-B™ capsule. The high adherence to the study protocol indicates that all the intervention types and routes of probiotic administration are acceptable. With the limitation that the overall sample size was small for this trial, Estring and RepHresh Pro-B use for 30 days did not lead to any serious adverse effects, with no clinical safety concerns identified in the trial. Reported AEs were largely mild, short-lived, and non-recurring. In addition, safety blood markers confirmed no clinically significant changes in health or immune cell count related to the intervention. Overall, our findings present comprehensive safety and feasibility data, highlighting the potential for future efficacy trials.

The AEs reported and the stability of blood markers align with the existing literature and were evenly distributed among participants across all four intervention types. In addition, self-reported adherence to daily use of each intervention over 30 days remained high and acceptable for the majority of participants. This corresponds to previous reports evaluating vaginal rings, along with oral and vaginal probiotics which have observed adherence rates >90% [31, 47–49]. Consistent with the safety profile of *Lactobacillus*-based probiotics including those reported specifically for *L. rhamnosus* GR-1 and *L. reuteri* RC-14, the most common local side effects were cramps/abdominal pain and vaginal discomfort which were mild and infrequent [25, 30, 31, 34, 50]. Similarly, the most common AEs included headaches and increased vaginal secretions for Estring© users [41, 51–53]. For the first time, we have recapitulated these findings for the vaginally administered RepHresh™ Pro-B™ product beyond five days and as a combined intervention with the Estring©. However, controversy regarding the existence and clinical significance of systemic effects of low-dose vaginal estrogens still exists [51]. Our results demonstrate no clinically significant changes in cholesterol, lipid profiles, triglycerides, liver enzymes, and blood cell count following 30 days of Estring© use. These findings align with studies among post-menopausal women which have shown that intravaginal estrogen results in minimal systemic absorption and normal circulating estradiol with no evidence of chronic disease risk [51–53]. When considering the potential application of these findings in Sub-Saharan Africa, where non-pill-based interventions are often preferred or oral treatments are not available or too expensive, our study offers valuable insights into feasible HIV prevention strategies [48, 54, 55].

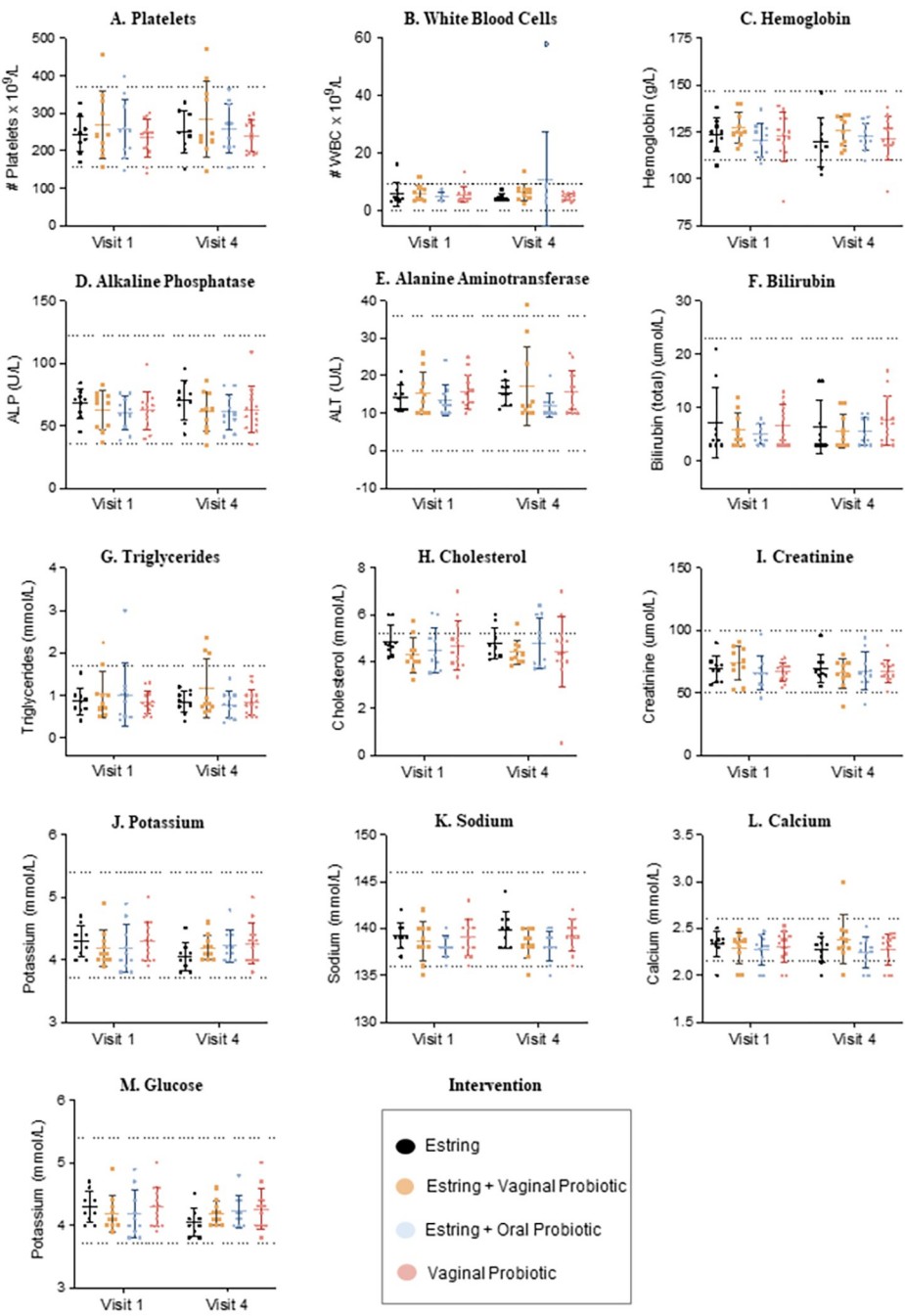

**Fig 5. Safety blood markers.** Blood collected at baseline (visit 2) and at the end of the intervention (visit 4) was used to measure metabolic, lipid and complete blood count panels pre and post intervention to evaluate safety. The safety blood marker panel consisted of 12 markers depicted by representative dot plots to monitor changes in health. Each data point is representative of a participant which has a paired sample taken at visit two and four. The four colours black, orange, blue, and red in each graph represent the treatment arm of each participant. In addition, the dotted lines in each graph indicate the clinically relevant thresholds representative of a healthy reference range for each safety marker. Overall, no clinically significant change was observed pre and post intervention suggesting that 30 days of the Estring© and/or probiotic RepHresh™ Pro-B™ does not induce changes to overall health. Moreover, the data is evenly distributed across intervention arms and thereby does not indicate an association to the type of intervention.

Formulation and delivery methods need to be considered when applying probiotics to different body sites and for diverse populations. By delivering local estrogen in conjunction with a probiotic we evaluated an innovative strategy to enhance and sustain the ability of *Lactobacillus* to restore a healthy microbiota in ACB women. The next step will be to determine if this leads to displacement of anaerobic and BV-associated microbiota, as well as enhancing host immunity, normalizing vaginal pH, and decreasing FGT inflammation along with clinical signs and symptoms of BV [5, 29, 37, 40]. It will also affirm whether *L. rhamnosus* and *L. reuteri*, may be more beneficial for ACB women where *L. crispatus* may not be the predominant microbial species within the VMB [27, 34, 56–58]. To date, there have been few clinical trials utilizing low-dose intravaginal estrogen with or without probiotics and studies have largely investigated post-menopausal women where results indicate estrogen, oral and vaginal probiotics are effective in reducing menopausal symptoms [41, 43, 59–62]. Consequently, the combined approach is hypothesized to confer multiple benefits to women of reproductive age including ACB women and those with recurrent BV or varying states of vaginal dysbiosis.

Our findings have confirmed the safety and feasibility across four intervention strategies. Subsequent studies will provide preliminary efficacy results according to changes in *Lactobacillus* dominance, microbial diversity, and inflammatory responses relative to baseline. A limitation of our study is that the adherence was calculated based primarily on self-reports contained in diaries completed by participants and verified with intervention products returned at the end of study, which does not rule out false reporting. While our results are promising, the generalizability is constrained by the study's small sample size and is considered the main limitation of this pilot phase 1 trial. Another limitation of this study is that we were unable to include a placebo arm and the study design relied on pre- and post- intervention data This has the inherent possibility of including confounding factors that temporally overlap with the intervention such as diet, seasonal and other changes. As such, careful consideration was given when interpreting results. Protocol adherence, which may rely on self-reports, was validated through additional measures such as PSA tests, and safety blood markers to minimize the impact of reporting bias [63]. Selection bias should also be considered as a potential limitation, given that participants were self-referred [64]. Future research should aim to replicate these findings in larger, more diverse cohorts with the inclusion of a placebo arm to validate and extend our understanding. Additionally, exploring the long-term efficacy of these interventions and their impact on the vaginal microbiome will be critical.

In conclusion, the findings from this phase 1 trial involving the administration of Estring© and RepHresh™ Pro-B™ demonstrated high enrollment, retention, and adherence, with no severe AEs reported across all four study groups. By delivering local estrogen in conjunction with a probiotic, we demonstrate a promising strategy to enhance and sustain the presence of *Lactobacillus* species and acidic pH in the vagina, potentially reducing HIV-1 risk among high-risk populations.

## Supporting information

**S1 Checklist. CONSORT checklist.**
(PDF)

**S1 File. Supplemental S1 and S2 Tables.**
(DOCX)

**S1 Protocol. Detailed study protocol.**
(DOCX)

## Acknowledgments

The authors would like to thank the CIHR HIV CTN Trial team including Judy Needham for CTN Project Management, Dana Nohynek for help with Regulatory Affairs, Melissa Babra and Nisha Shewaramani for Data Management, Terry Lee for statistical analysis and Leslie Love for Study monitoring.

## Author Contributions

**Conceptualization:** Jocelyn M. Wessels, Gregor Reid, Wangari Tharao, Fiona Smaill, Charu Kaushic.

**Data curation:** Biban Gill, Christina L. Hayes, Jenna Ratcliffe, Elizabeth Ball.

**Formal analysis:** Biban Gill, Jocelyn M. Wessels, Christina L. Hayes, Elizabeth Ball, Fiona Smaill, Charu Kaushic.

**Funding acquisition:** Fiona Smaill, Charu Kaushic.

**Investigation:** Christina L. Hayes, Jenna Ratcliffe, Junic Wokuri.

**Methodology:** Biban Gill, Jocelyn M. Wessels, Christina L. Hayes, Jenna Ratcliffe, Junic Wokuri, Fiona Smaill, Charu Kaushic.

**Project administration:** Jenna Ratcliffe, Wangari Tharao, Charu Kaushic.

**Resources:** Wangari Tharao, Charu Kaushic.

**Software:** Biban Gill, Christina L. Hayes, Elizabeth Ball.

**Supervision:** Rupert Kaul, Jesleen Rana, Muna Alkhaifi, Fiona Smaill, Charu Kaushic.

**Validation:** Biban Gill, Jocelyn M. Wessels, Christina L. Hayes, Jenna Ratcliffe, Elizabeth Ball.

**Visualization:** Biban Gill, Jocelyn M. Wessels, Christina L. Hayes.

**Writing – original draft:** Biban Gill, Elizabeth Ball, Fiona Smaill, Charu Kaushic.

**Writing – review & editing:** Biban Gill, Jocelyn M. Wessels, Christina L. Hayes, Jenna Ratcliffe, Junic Wokuri, Gregor Reid, Rupert Kaul, Jesleen Rana, Muna Alkhaifi, Wangari Tharao, Fiona Smaill, Charu Kaushic.

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
