## [Decision Letter · Decision Letter 0]

26 Jan 2024

PONE-D-23-43948Feasibility, safety and tolerability of estrogen and/or probiotics for improving vaginal health in Canadian African, Caribbean, and Black women: A pilot phase 1 clinical trialPLOS ONE

Dear Dr. Kaushic

Thank you for submitting your manuscript to PLOS ONE. After careful consideration, we feel that it has merit but does not fully meet PLOS ONE’s publication criteria as it currently stands. Therefore, we invite you to submit a revised version of the manuscript that addresses the points raised during the review process.

**Please address commments of both reviewers**Please ensure that your decision is justified on PLOS ONE’s publication criteria and not, for example, on novelty or perceived impact.

We look forward to receiving your revised manuscript.

Kind regards,

Alan Landay

Academic Editor

PLOS ONE

Journal Requirements:

“This study was funded by CIHR Grants FRN#159229 and FRN#154047 (CK) and in kind contributions from CIHR HIV Clinical Trials Network (FS, CK).

B.G was partially supported by a Post-doctoral Award from CIHR FRN#181875”

“I have read the journal's policy and the authors of this manuscript have the following competing interests: GR developed the probiotic strains GR-1 and RC-14 but has had no financial interest in them for 15 years. GR consults for Seed, a company producing probiotic strains not used in this study. All the other authors have declared no conflict of interest.”

We note that one or more of the authors are employed by a commercial company: Seed

“This study was supported by CIHR Grants FRN# 159229 and #126019 (C.K.) and by in kind support from the CIHR Canadian HIV Trials Network (CTN 308). B.G was partially supported by a Post-doctoral Award from CIHR FRN#181875.”

“This study was funded by CIHR Grants FRN#159229 and FRN#154047 (CK) and in kind contributions from CIHR HIV Clinical Trials Network (FS, CK).

B.G was partially supported by a Post-doctoral Award from CIHR FRN#181875”

Additional Editor Comments (if provided):

Please address the comments of the reviewers

Reviewers' comments:

Reviewer's Responses to Questions

**Comments to the Author**

1. Is the manuscript technically sound, and do the data support the conclusions?

Reviewer #1: Yes

Reviewer #2: Yes

2. Has the statistical analysis been performed appropriately and rigorously? 

Reviewer #1: Yes

Reviewer #2: Yes

3. Have the authors made all data underlying the findings in their manuscript fully available?

Reviewer #1: Yes

Reviewer #2: Yes

4. Is the manuscript presented in an intelligible fashion and written in standard English?

Reviewer #1: Yes

Reviewer #2: Yes

5. Review Comments to the Author

Reviewer #1: This paper reports the results of a prospective, randomized, open-label phase 1 clinical trial to determine the feasibility, safety, and tolerability of administering low-dose estrogen, probiotics, or both in combination to improve vaginal health and decrease HIV susceptibility. The manuscript if well written, easy to follow and straight forward. I am particularly pleased with the decision to focus enrollment on African, Caribbean, and Black women, who are historically underrepresented in research. I wholeheartedly agree that acceptance of formulation and delivery methods need to be considered when applying probiotics to different body sites and for diverse populations. I was a bit disappointed about the lack of biological outcome data in this paper, like for example displacement of anaerobic and BV-associated microbiota, normalizing vaginal pH, and decreasing genital inflammation along with clinical signs and symptoms of BV (it looks like this part will be published separately in the future). Unless there is a rationale for splitting the paper, I would prefer to see a complete story (this paper feels amputated). Other limitations include the small sample size limiting generalization (was acknowledged by the authors), lack of placebo arm (should be acknowledged by the authors). I was also surprised by the number of adverse events. Are those severe systemic effects (like nausea, light headedness, headache, cramps) expected with this type of intervention? It is hard to evaluate without a placebo arm.

Importantly, do NOT use the word subject when referring to study participants (it is stigmatizing and not acceptable).

Reviewer #2: This paper summarizes the safety, feasibility, and tolerability of innovative intervention regimens to improve vaginal health. It was a randomized phase 1 open label in which participants were randomized to one of 4 arms: Intravaginal estradiol (estring 7.5mg/day), vaginal probiotic 2x/day, estradiol + vaginal, estradiol + oral twice daily. Interventions were taken for 30 days. There was high and comparable adherence across arms. While the AE rate was high, they were primarily mild and resolved by end of study, with no SAEs. The results are sufficient to move forward with larger trials of these potential interventions. The paper is nicely written, and only minor clarifications are requested.

Abstract: Concise and well-written.

Results : could the authors please clarify what is the unit on adherence? E.g., is 0.93  93% of doses were taken?

Introduction

- Well written and generally sufficient. There seems to be a jump in rationale explaining the selection of Lacticacseibacillus rhamnosus and Limosillactobacillus reuteri (lines 92-97). Could the authors please add a sentence or two about how these strains address the sentence (92-94), or the “properties” they are addressing in line 94? Reviewer notes later there are some mechanisms at lines 111-113 that may flow better above.

- Lines 106-108, could authors clarify here whether the studies citing topical estrogen creams employed safely and well-tolerated were in pre-menopausal women? It may be there in the citations, but just a few words to clarify and save the reader from having to look something up.

- Line 109. The use of the word “demonstrate” is confusing in this sentence: “we are testing an innovative strategy”?

Materials and Methods

- Line 135. Could the authors please specify the amount of compensation?

- Lines 145-146. Could the authors please explain at least some of the inclusion and exclusion criteria in the text? To have none there is disruptive to reading. Or make this a table in the main paper (I don’t think Plos limits these?) because it’s disruptive to have to go to supplemental to read this.

o Irregular menstrual cycle is listed as an exclusion criterion. Could the authors please explain how this was assessed?

- Explanation of randomization process is clear.

- Description of study visits and procedures are clear.

Results

- The CONSORT flow diagram is clear and nicely done.

o Could the authors please explain why 4 participants withdrew consent?

- Lines 227-229: Reviewer is confused on what is meant by “prematurely terminated the study” (dropped out? Stopped coming for study visits?) as they are then “considered non-adherent” (line 229) so it would be important to understand why they were non-adherent/prematurely terminated.

- Line 239, it is unclear why “congenital anomaly” is listed as a potential SAE of this intervention?

- Lines 288-290: Could the authors please clarify the adherence rate units – is tis 92% 93% 97% etc.? Are these adherence rates for all 50 women who were randomized or just those n=41 completing? It seems like it is for all based implication in lines 292-293. But this is somewhat confusing b/c the CONSORT diagram indicates that these “participants terminating study prematurely” (line 293) are not included in analyses. But then the safety markers are also analyzed across the 50 participants (368)

- Kudos to investigators on the high adherence and completion rates! Were participants provided any incentives or reminders?

Discussion

- Well written and concise. Hits the main points.

- As with comment above, wording in lines 416-418 is confusing use of “demonstrate” – “we demonstrate an innovative strategy”. “we evaluate? Test? Examine?”

- Lines 421-422 – it seems something got cut out of this sentence. In what ways “more beneficial” for?

6. PLOS authors have the option to publish the peer review history of their article (what does this mean?). If published, this will include your full peer review and any attached files.

Reviewer #1: No

Reviewer #2: No

---

## [Author Response · Author response to Decision Letter 0]

6 Mar 2024

Reply to Reviewers:

Additional Editor Comments (if provided):

Please address the comments of the reviewers

We thank the Editor for their comment and have addressed all the comments made by reviewers as detailed below.

Reviewer #1:

This paper reports the results of a prospective, randomized, open-label phase 1 clinical trial to determine the feasibility, safety, and tolerability of administering low-dose estrogen, probiotics, or both in combination to improve vaginal health and decrease HIV susceptibility. The manuscript if well written, easy to follow and straight forward. I am particularly pleased with the decision to focus enrollment on African, Caribbean, and Black women, who are historically underrepresented in research. I wholeheartedly agree that acceptance of formulation and delivery methods need to be considered when applying probiotics to different body sites and for diverse populations.

Comments to address:

1) I was a bit disappointed about the lack of biological outcome data in this paper, like for example displacement of anaerobic and BV-associated microbiota, normalizing vaginal pH, and decreasing genital inflammation along with clinical signs and symptoms of BV (it looks like this part will be published separately in the future). Unless there is a rationale for splitting the paper, I would prefer to see a complete story (this paper feels amputated).

We thank the reviewer for their comments and agree that the biological outcome data is an important aspect of the study. We are in the process of completing this analysis which will include biological outcomes related to the results of the microbiome analysis and the impact of the intervention on genital inflammation. Given the extensive biological data including microbiome analysis, correlation of treatment benefits with microbiome analysis, qPCR and multi-plex cytokine analysis and their correlation to clinical benefits that was conducted, we felt it best to separate the two papers to ensure an in-depth coverage of safety, feasibility and tolerability as well as biological response. The correlation analysis to synthesize clinical benefit is currently being completed, therefore it would be best to publish this first paper first. Combining both sets of data will make the manuscript too lengthy and delay communicating the information on primary outcomes of the Phase I clinical trial.

2) Other limitations include the small sample size limiting generalization (was acknowledged by the authors), lack of placebo arm (should be acknowledged by the authors).

We thank the reviewer for their comment and while this phase 1 trial was designed to test the effect of intervention on each participant before and after the treatment, we acknowledge that the inclusion of a placebo arm would be a better study design. We acknowledge that the conclusion of the effect of the intervention in this study design could be confounded by other temporal changes that overlap with the length of intervention, such as diet, seasonal and other changes. The fourth paragraph of the discussion (lines 450-453) has been updated to acknowledge the lack of placebo arm as a limitation due to the availability of resources. In addition, we have made note of the inclusion of a placebo arm for future studies. 

3) I was also surprised by the number of adverse events. Are those severe systemic effects (like nausea, light headedness, headache, cramps) expected with this type of intervention? It is hard to evaluate without a placebo arm.

We appreciate the reviewer’s insights and concern. The adverse events reported by participants including nausea, light headedness, headache, and cramps were not surprising with this type of intervention, since these are listed as side effects of RepHresh™ Pro-B™ and/or Estring©. The study protocol (Protocol S1) included with the supplemental materials provides a detailed overview of the expected adverse events based on previous studies and manufacturer details. The second paragraph (Lines 418-421) of the discussion quoted below, highlights these adverse events as the most commonly reported side effects for these study products, with multiple references included. “Consistent with the safety profile of Lactobacillus-based probiotics including those reported specifically for L. rhamnosus GR-1 and L. reuteri RC-14, the most common local side effects were cramps/abdominal pain and vaginal discomfort which were mild and infrequent (25,30,31,34,50). Similarly, the most common AEs included headaches and increased vaginal secretions for Estring© users (41,51-53).” Of note, these effects were temporary and almost all resolved within a short time. Importantly the side effects did not have any impact on the acceptability and adherence of the intervention.

4) Importantly, do NOT use the word subject when referring to study participants (it is stigmatizing and not acceptable).

We thank the reviewer for highlighting this issue. We agree that participant is much better terminology than subjects and apologies for this oversight. The manuscript has been updated accordingly and no longer uses the term “subject(s)” to refer to study participants. The four instances in which this was done have now been changed to “participant(s)”, shown in track changes in the revised manuscript. 

Reviewer #2:

This paper summarizes the safety, feasibility, and tolerability of innovative intervention regimens to improve vaginal health. It was a randomized phase 1 open label in which participants were randomized to one of 4 arms: Intravaginal estradiol (estring 7.5mg/day), vaginal probiotic 2x/day, estradiol + vaginal, estradiol + oral twice daily. Interventions were taken for 30 days. There was high and comparable adherence across arms. While the AE rate was high, they were primarily mild and resolved by end of study, with no SAEs. The results are sufficient to move forward with larger trials of these potential interventions. The paper is nicely written, and only minor clarifications are requested.

Comments to address:

1) Could the authors please clarify what is the unit on adherence? E.g., is 0.93  93% of doses were taken?

We thank the reviewer for their comment regarding the adherence units. We have updated the manuscript to list adherence metrics as a percentage to improve readability. The specific definitions describing how adherence values were determined are described in the “Outcome Measures” section in the Materials and Methods (lines 238-240). 

2) There seems to be a jump in rationale explaining the selection of Lacticacseibacillus rhamnosus and Limosillactobacillus reuteri (lines 92-97). Could the authors please add a sentence or two about how these strains address the sentence (92-94), or the “properties” they are addressing in line 94? Reviewer notes later there are some mechanisms at lines 111-113 that may flow better above.

We thank the reviewer for their feedback. We have updated the introduction to more clearly address the properties mentioned on line 92-94 by taking the reviewer’s suggestion and moving the description of the mechanisms on from next paragrpah to add to previous paragraph (lines 91-96) to improve the flow of this section.

3) Lines 106-108, could authors clarify here whether the studies citing topical estrogen creams employed safely and well-tolerated were in pre-menopausal women? It may be there in the citations, but just a few words to clarify and save the reader from having to look something up.

We thank the reviewer for their insight and have included a statement clarifying that the studies cited in the text for both topical creams and vaginal rings containing estrogen were in post-menopausal women, to improve clarity of this section of the manuscript (lines114-116). 

4) Line 109. The use of the word “demonstrate” is confusing in this sentence: “we are testing an innovative strategy”?

We thank reviewer 2 for their feedback and have taken their suggestion to replace the word “demonstrate” on line 109 with “we are testing an innovative strategy” to improve clarity of this sentence (line 117).

5) Line 135. Could the authors please specify the amount of compensation?

We thank reviewer 2 for their comment and have included the amount of compensation ($100/visit) in the Material and Methods. Additional details regarding the incentive referral process can be found in the detailed study protocol (Protocol-S1) which is included as a supplemental document.

6) Lines 145-146. Could the authors please explain at least some of the inclusion and exclusion criteria in the text? To have none there is disruptive to reading. Or make this a table in the main paper (I don’t think Plos limits these?) because it’s disruptive to have to go to supplemental to read this.

We appreciate reviewer 2’s comment and have moved the inclusion and exclusion criteria table from the supplemental information to the main text. This is now listed as Table 1 in the manuscript with a detailed description of the criteria described in the study protocol (S1-Protocol) attached as supplemental material.

7) Irregular menstrual cycle is listed as an exclusion criterion. Could the authors please explain how this was assessed?

Thank you for this comment. The irregular menstrual cycle was assessed by the study nurse during the initial screening interview. This is described in the study protocol (Protocol S1) included as supplemental materials. We have updated the methods section as described in the previous comment to now include inclusion/exclusion criteria in Table 1, with a clear reference to the study protocol for further details.

8) Could the authors please explain why 4 participants withdrew consent?

We appreciate reviewer 2 inquiring about the participants who withdrew consent. These four participants no longer wished to partake in the study and chose to withdraw between visits 2 and 3. They explained to the study nurse that they had not been able to use the intervention as required in the study protocol and therefore wished to withdraw from the study. 

9) Lines 227-229: Reviewer is confused on what is meant by “prematurely terminated the study” (dropped out? Stopped coming for study visits?) as they are then “considered non-adherent” (line 229) so it would be important to understand why they were non-adherent/prematurely terminated.

We thank reviewer 2 for their comment and apologize for the confusion. We have now clarified the three categories of participants who were given a value of 0 in the adherence metrics: 1) those who were non-adherent and wished to withdraw consent to participate in the study (n=4) 2) those who were lost to follow up (n=1) and 3) those who were non-adherent (but did not withdraw consent) and did not complete the study (n=4). As described in the following line (lines 240-243) participants who did not complete the intervention protocol (for any of the three reasons above) were assigned a value of 0 when calculating adherence metrics. We have removed the text and removed “premature termination” as it was creating confusion.

10) Line 239, it is unclear why “congenital anomaly” is listed as a potential SAE of this intervention?

We thank reviewer 2 for their comment. The inclusion of “congenital anomaly” listed as an SAE is based on the definition of Severe Adverse Events. This is a general, all-encompassing definition of SAEs for clinical trials, as per US Department of HHS Guidelines for Grading Severity of Adult and Pediatric Adverse Events and is not specific to this trial. We have referenced the source for this definition in the text on Line 255. 

11) Lines 288-290: Could the authors please clarify the adherence rate units – is tis 92% 93% 97% etc.? Are these adherence rates for all 50 women who were randomized or just those n=41 completing? It seems like it is for all based implication in lines 292-293. But this is somewhat confusing b/c the CONSORT diagram indicates that these “participants terminating study prematurely” (line 293) are not included in analyses. But then the safety markers are also analyzed across the 50 participants (368)

We thank reviewer 2 for their comment. The adherence rate units are indeed same as % and have been altered in the revised manuscript and associated Figure 5. Reviewer 2 is correct that the adherence rates are determined based on all 50 women who received the intervention. The text on Lines 304-308 has been updated to better describe how adherence was determined and which participants were included in the calculation. 

12) Kudos to investigators on the high adherence and completion rates! Were participants provided any incentives or reminders?

We thank reviewer 2 for their positive feedback. Participants were provided with compensation for their time and additional expenses and the nurse sent text/phone call reminders of clinic appointments. The details for the compensation/incentives can be found in the study protocol (Protocol S1).

13) As with comment above, wording in lines 416-418 is confusing use of “demonstrate” – “we demonstrate an innovative strategy”. “we evaluate? Test? Examine?”

We thank reviewer 2 for their comment and have updated the wording to avoid the use of the word “demonstrate” in this sentence. We have replaced “demonstrate” with “evaluate” to improve clarity and readability (Line 434).

14) Lines 421-422 – it seems something got cut out of this sentence. In what ways “more beneficial” for?

We thank reviewer 2 for making note of this oversight. We have updated this sentence to reflect that the “intervention may be more beneficial for ACB women where L. crispatus may not be the predominant microbial species present within the vaginal microbiome”. (Lines 439-440)

---

## [Decision Letter · Decision Letter 1]

29 May 2024

PONE-D-23-43948R1Feasibility, safety and tolerability of estrogen and/or probiotics for improving vaginal health in Canadian African, Caribbean, and Black women: A pilot phase 1 clinical trialPLOS ONE

Dear Dr. Kaushic,

Thank you for submitting your manuscript to PLOS ONE. After careful consideration, we feel that it has merit but does not fully meet PLOS ONE’s publication criteria as it currently stands. Therefore, we invite you to submit a revised version of the manuscript that addresses the points raised during the review process.

We look forward to receiving your revised manuscript.

Kind regards,

Alan Landay

Academic Editor

PLOS ONE

Journal Requirements:

Reviewers' comments:

Reviewer's Responses to Questions

**Comments to the Author**

1. If the authors have adequately addressed your comments raised in a previous round of review and you feel that this manuscript is now acceptable for publication, you may indicate that here to bypass the “Comments to the Author” section, enter your conflict of interest statement in the “Confidential to Editor” section, and submit your "Accept" recommendation.

Reviewer #3: (No Response)

2. Is the manuscript technically sound, and do the data support the conclusions?

Reviewer #3: Yes

3. Has the statistical analysis been performed appropriately and rigorously? 

Reviewer #3: Yes

4. Have the authors made all data underlying the findings in their manuscript fully available?

Reviewer #3: Yes

5. Is the manuscript presented in an intelligible fashion and written in standard English?

Reviewer #3: Yes

6. Review Comments to the Author

Reviewer #3: Comments:

Statistically, this manuscript is pretty straightforward. For phase 1 safety trials, this is usually the case since the purpose and sample size are not conducive for performing more complicated analyses. Your stated goals of the trial in line 114 align with this and, as I see it, assessing "feasibility, safety, and tolerability" are more about estimating proportions and percentages than doing any sort of comparative analyses.

That said, I am not sure about the sample sizes for the arms in this trial. I know there exists literature on calculating the sample size for phase 1 dose-finding trials, but I know less about safety trials. Logically, I would think that the sample size should be large enough to capture AEs (and especially SAEs) that occur with some frequency in the population being studied. For the overall numbers, I think you are ok. That's because, if we assume that the probability of an AE is 0.1, then, using the binomial distribution, the probability that you would not detect that AE in 41 people is 0.01. So, only a 1% chance that it would be missed. On the other hand, if we use a sample size of 13 people, that jumps to 0.25 (and 0.31 for 11 people). That's a pretty high probability and I think is too high for comfort. There are no power analyses reported in this manuscript. I think that is a negative, but that can't be changed retrospectively.

I am struggling the most with PLOS publication criterion number 3 (https://journals.plos.org/plosone/s/criteria-for-publication) since I think the product-specific sample sizes are too low to be robust. I think the conclusions inferred by product, such as lines 384-386, are too strong given the small sample size. I think a greater focus on the overall results and softening the language around the product-specific conclusions is needed.

Specific comments:

1. (lines 146-9) Please reference the computer program used for randomization.

2. (Table 4) I believe the overall sample size for Visit 5 is a typo and should read 41 instead of 4.

7. PLOS authors have the option to publish the peer review history of their article (what does this mean?). If published, this will include your full peer review and any attached files.

Reviewer #3: No

---

## [Author Response · Author response to Decision Letter 1]

12 Jun 2024

We thank the reviewer for this insightful comment and appreciate the opportunity to clarify the details of our sample sizes within the intervention arms. 

This study was designed as a pilot Phase I clinical trial study with the purpose was to examine the feasibility of recruitment, randomization, retention, assessment procedures, and implementation of an novel intervention. 

The reviewer questions how to approach calculating the sample size for a Phase 1 safety study. A pilot study cannot inform the safety testing of a phase 1 study and no sample size calculations based on safety outcomes are recommended. (10.1016/j.jpsychires.2010.10.008). Only in an extreme, unfortunate case, where a death occurs or repeated serious adverse events surface, can a pilot study inform the safety of testing an intervention due to the small sample size. However, pilot studies provide an opportunity to implement and examine the feasibility of the adverse event reporting system.

While originally each intervention arm was planned to be 20 participants, the trial was stopped after two years when the goals for feasibility and safety data were met, due to the multiple starts and stops in the trial due to COVID. Given the smaller number in each arm of the intervention, we did not statistical power to analyze each study arm. Instead the AE incidence for the Estring was examined by combining all 3 groups that received this intervention (Estring alone, Estring + oral probiotics, Estring+ vaginal probiotics) and for probiotics also from 3 groups (Vaginal probiotic, Estring + oral probiotics, Estring+ vaginal probiotics). For the Estring intervention arm, the product-specific sample size was 37 at enrollment and 28 at completion, while for the probiotics arm, it was 37 at enrollment and 32 at completion. This adjustment brings the probability of missing an AE closer to 2-5% based on probability of AE as 0.1, as suggested by the reviewer, rather than 25-30%. Additionally, we want to reiterate that despite the smaller sample sizes, there were no signals for any serious adverse events throughout the entirety of our study. Nevertheless, we acknowledge the importance of exercising caution and have reflected this sentiment in the revised language in the Discussion section, where we state “With the limitation that the overall sample size was small for this trial, Estring and RepHresh Pro-B use for 30 days did not lead to any serious adverse effects, with no clinical safety concerns identified in the trial” 

We sincerely appreciate the reviewer’s valuable feedback, which has allowed us to provide a more nuanced discussion of our findings and their implications.

Reviewer: Specific comments:

1. (lines 146-9) Please reference the computer program used for randomization.

REPLY: The acquisition of test products, packaging, randomization were contracted out to BARL (Bay Area Research Logistics) who specialize in clinical supply and logistical management of clinical trials. The randomization code was generated by the statistician in BARL in block sizes of 4 or 8. 

2. (Table 4) I believe the overall sample size for Visit 5 is a typo and should read 41 instead of 4.

REPLY: Many thanks to the reviewer for catching this oversight. This has been corrected.

---

## [Decision Letter · Decision Letter 2]

6 Sep 2024

PONE-D-23-43948R2Feasibility, safety and tolerability of estrogen and/or probiotics for improving vaginal health in Canadian African, Caribbean, and Black women: A pilot phase 1 clinical trialPLOS ONE

Dear Dr. Kaushic,

Thank you for submitting your manuscript to PLOS ONE. After careful consideration, we feel that it has merit but does not fully meet PLOS ONE’s publication criteria as it currently stands. Therefore, we invite you to submit a revised version of the manuscript that addresses the points raised during the review process.

We look forward to receiving your revised manuscript.

Kind regards,

Kazunori Nagasaka

Academic Editor

PLOS ONE

Journal Requirements:

Additional Editor Comments:

Dear Authors,

Thank you for submitting your research in Plos One.

Please revise the manuscript according to the reviewers comments.

We look forward to your revised manuscript.

Sincerely,

Kazunkori Nagasaka

Reviewers' comments:

Reviewer's Responses to Questions

**Comments to the Author**

1. If the authors have adequately addressed your comments raised in a previous round of review and you feel that this manuscript is now acceptable for publication, you may indicate that here to bypass the “Comments to the Author” section, enter your conflict of interest statement in the “Confidential to Editor” section, and submit your "Accept" recommendation.

Reviewer #3: (No Response)

Reviewer #4: All comments have been addressed

Reviewer #5: (No Response)

2. Is the manuscript technically sound, and do the data support the conclusions?

Reviewer #3: No

Reviewer #4: Yes

Reviewer #5: Yes

3. Has the statistical analysis been performed appropriately and rigorously? 

Reviewer #3: No

Reviewer #4: Yes

Reviewer #5: Yes

4. Have the authors made all data underlying the findings in their manuscript fully available?

Reviewer #3: Yes

Reviewer #4: Yes

Reviewer #5: Yes

5. Is the manuscript presented in an intelligible fashion and written in standard English?

Reviewer #3: Yes

Reviewer #4: Yes

Reviewer #5: Yes

6. Review Comments to the Author

Reviewer #3: Thank you for your thorough consideration of my comments. That said, I am still coming to the same conclusions as my previous review. Based on your response, specifically this:

"This study was designed as a pilot Phase I clinical trial study with the purpose was to examine the feasibility of recruitment, randomization, retention, assessment procedures, and implementation of an novel intervention."

safety is not a part of this study's purpose. And yet the title and abstract seem to report safety findings. Demonstration projects are important and I can understand the argument that such a project is needed to assess the feasibility of testing an intervention and data collection systems. But, the way this manuscript is structured it still looks heavily like you are reporting the results of a safety trial and that safety is one of the objectives of this manuscript.

I think it is unreasonable to expect a phase 1 trial to capture an extremely rare event, such as a 1:1000 event. But, I think it is reasonable for a phase 1 trial to capture a 1:100, which I don't see as an extreme case.

Thus, I still feel the sample sizes are not large enough to produce robust results and that criterion #3 (https://journals.plos.org/plosone/s/criteria-for-publication) has not been met.

Reviewer #4: The authors have addressed all issues, and I believe the manuscript is ready for publication in its current form.

Reviewer #5: This is an important manuscript addressing a key research question – understanding and mitigating modifiable risk factors for HIV transmission in a high-risk population. The introduction adequately lays out the scientific rationale for the feasibility trial and underlying clinical question being discussed, including current data gaps – need for research to better prevent/treat BV in order to reduce biological susceptibility to HIV, and lack of safety data on use of oral probiotics combined with estrogen vaginally for longer than 5 days. The primary and secondary outcomes are clearly defined, and the study design which includes randomization and intention to treat analysis is sound. The methods section is well laid out and use a schedule events make it easy to follow.

A few thing can be further clarified to strengthen the interpretation of results and support the study’s conclusions:

1) Adherence assessment:

- How was adherence measured exactly, in the participants who completed the trial?

- The manuscript states ‘adherence metrics were determined according to either the ratio of probiotics consumed relative to the total dispensed, or the number of days the estring was used relative to the study duration.” – was this all based on self-report, or were there any objective measures of assessing adherence? The manuscript states that participants kept a diary – what exactly did they note? E.g., those using a probiotics twice a day, did they indicate date/time each probiotic was used for the full 30 days, and how was this assessed/graded to give an adherence score of close to 1 (perfect adherence)? It is often difficult for patients to take a daily pill for 30 days, and we expect it would be difficult to do so twice daily for 30 days, so how was this assessed to give the high adherence scores noted? This needs to be justified further

- Can an addendum of a sample study diary that is filled out be included in the appendix to show an example of ‘good’ adherence (assuming filled out daily), vs non-ideal adherence? What data did the authors use to back the statement “diary completion across all visits and interventions was 100% among those who completed the study,” and can the authors more explicitly address how this method of assessing diary completion gives them confidence that the participants used the capsules twice daily, which then strengthens the safety conclusions of the trial? This is important, because non-optimal adherence affects the safety outcomes which are important for the trial. If the authors cannot confidently state that the intervention was not used twice daily for 30 days, then this should be included in the limitations to inform interpretation of the AEs noted.

2) Adverse event evaluation

- What scale was used to grade the AEs, and is it a standardized scale/one used previously in similar studies? What differentiated a mild from moderate AE? If a standardized grading scale was not used, consider noting this as a limitation as it may reduce reproducibility

3) Completion of sexual history questionnaire as adherence assessments

- Can the authors please clarify how completion of a the sexual history questionnaire is used as an adherence assessment method? What is the rationale for this? Has this been validated previously or used in similar studies?

7. PLOS authors have the option to publish the peer review history of their article (what does this mean?). If published, this will include your full peer review and any attached files.

Reviewer #3: No

Reviewer #4: No

Reviewer #5: No

---

## [Author Response · Author response to Decision Letter 2]

4 Oct 2024

Reviewer #3: Thank you for your thorough consideration of my comments. That said, I am still coming to the same conclusions as my previous review. Based on your response, specifically this:

"This study was designed as a pilot Phase I clinical trial study with the purpose was to examine the feasibility of recruitment, randomization, retention, assessment procedures, and implementation of an novel intervention."

safety is not a part of this study's purpose. And yet the title and abstract seem to report safety findings. Demonstration projects are important and I can understand the argument that such a project is needed to assess the feasibility of testing an intervention and data collection systems. But, the way this manuscript is structured it still looks heavily like you are reporting the results of a safety trial and that safety is one of the objectives of this manuscript.

I think it is unreasonable to expect a phase 1 trial to capture an extremely rare event, such as a 1:1000 event. But, I think it is reasonable for a phase 1 trial to capture a 1:100, which I don't see as an extreme case.

Thus, I still feel the sample sizes are not large enough to produce robust results and that criterion #3 (https://journals.plos.org/plosone/s/criteria-for-publication) has not been met.

REPLY: We are sorry that the reviewer feels that the manuscript does not meet the criteria for reporting safety. As explained in our previous reply, we did calculate the probability of missing an AE based on the 1:100 (0.01) chance that the reviewer said was acceptable and we showed that based on the sample size of the study, the probability of missing an AE of 1:100 was between 2-5% and not 25-30% as projected by the reviewer. Furthermore, we altered the language in the Discussion and added the language that “With the limitation that the overall sample size was small for this trial, Estring and RepHresh Pro-B use for 30 days did not lead to any serious adverse effects, with no clinical safety concerns identified in the trial” . We have now added “2-5% chance of missing 1:100 AE, based on sample size” to the discussion to be transparent about the limitation of safety data based on the sample size in this trial (Page 21, line 386)

REPLY: 

Reviewer #4: The authors have addressed all issues, and I believe the manuscript is ready for publication in its current form.

REPLY: We thank the reviewer and really appreciate their comment that the manuscript is ready for publication. 

Reviewer #5: This is an important manuscript addressing a key research question – understanding and mitigating modifiable risk factors for HIV transmission in a high-risk population. The introduction adequately lays out the scientific rationale for the feasibility trial and underlying clinical question being discussed, including current data gaps – need for research to better prevent/treat BV in order to reduce biological susceptibility to HIV, and lack of safety data on use of oral probiotics combined with estrogen vaginally for longer than 5 days. The primary and secondary outcomes are clearly defined, and the study design which includes randomization and intention to treat analysis is sound. The methods section is well laid out and use a schedule events make it easy to follow.

A few thing can be further clarified to strengthen the interpretation of results and support the study’s conclusions:

1) Adherence assessment:

- How was adherence measured exactly, in the participants who completed the trial?

- The manuscript states ‘adherence metrics were determined according to either the ratio of probiotics consumed relative to the total dispensed, or the number of days the estring was used relative to the study duration.” – was this all based on self-report, or were there any objective measures of assessing adherence? The manuscript states that participants kept a diary – what exactly did they note? E.g., those using a probiotics twice a day, did they indicate date/time each probiotic was used for the full 30 days, and how was this assessed/graded to give an adherence score of close to 1 (perfect adherence)? It is often difficult for patients to take a daily pill for 30 days, and we expect it would be difficult to do so twice daily for 30 days, so how was this assessed to give the high adherence scores noted? This needs to be justified further

REPLY: Thank you for your comments. In this study, adherence was measured primarily through self-report, as participants maintained a daily diary throughout the trial and secondly by asking the participants to return of unused probiotic capsules and vaginal ring at the end of the study. Specifically, for participants using probiotics, they recorded each dose taken, including the date and time of day (morning and evening), in their diaries. The participants brought the diary with them to the clinic during their visits which was reviewed by the nurse at Visit 3 (Day 14) and Visit 4 (final visit for end of study, Day 30). At the final visit, the diaries, unused study products (Estring and probiotic pills) as well as empty containers were returned to the clinic. Adherence scores were calculated based on these self-reports by comparing the number of doses recorded in the diary against the total number of doses expected to be used over the study period. The number of pills returned was counted to match with diary reports. Regarding the high adherence scores, while we acknowledge that adherence to twice-daily probiotics for 30 days can be challenging, the data were derived from participant-reported diaries, which were reviewed and cross-checked with the number of probiotics dispensed/returned. This method is commonly used for microbicide and probiotic trials (Cohen et al, 2020 NEJM (ref 31), Happel et al, BMC Infec Dis 2020 (ref 47), Montogomery et al, 2012, AIDS Behav (ref 48). However, we recognize that reliance on self-report can have limitations, such as false reporting. This limitation has been further emphasized in the discussion (Page 23, line 427-30).

- Can an addendum of a sample study diary that is filled out be included in the appendix to show an example of ‘good’ adherence (assuming filled out daily), vs non-ideal adherence? What data did the authors use to back the statement “diary completion across all visits and interventions was 100% among those who completed the study,” and can the authors more explicitly address how this method of assessing diary completion gives them confidence that the participants used the capsules twice daily, which then strengthens the safety conclusions of the trial? This is important, because non-optimal adherence affects the safety outcomes which are important for the trial. If the authors cannot confidently state that the intervention was not used twice daily for 30 days, then this should be included in the limitations to inform interpretation of the AEs noted.

REPLY: Thank you for this comment. The diary completion of 100% refers to number of participants who completed the study and returned completed diary (97% including those that discontinued the intervention). It does not imply that there was 100% adherence to the intervention. Once the diaries were returned, the data from the diaries were counted and adherence rate was calculated across different intervention groups and across the whole study. Based on these calculations, the adherence met the 80% threshold we wanted to see in the trial. We have added some example pages of participants in Appendix 1 to show what high and non-optimal adherence looked like. 

Data was collected and tabulated from all diaries and adherence was calculated for each intervention group. Overall, the adherence for Estring from diary data collected from groups 1, 2 and 3 was 97% and probiotic from groups 2, 3 and 4 was 92% (Fig 3 and Table 4). Given the high adherence rate, barring false self reports, we do not think this would affect the safety outcomes.

2) Adverse event evaluation

- What scale was used to grade the AEs, and is it a standardized scale/one used previously in similar studies? What differentiated a mild from moderate AE? If a standardized grading scale was not used, consider noting this as a limitation as it may reduce reproducibility

REPLY: Adverse events were categorized and the intensity for each AE graded according to the guidelines provided by the Division of AIDS (DAIDS, NIH) Adverse Event Grading Table which has been used in numerous clinical trials funded by DAIDS for safety data reporting to maintain accuracy and consistency in the evaluation of AEs. Table for Grading the Severity of Adult and Pediatric Adverse Events V2.1 (https://rsc.tech-res.com/clinical-research-sites/safety-reporting/daids-grading-tables ). This was cited in the Outcome measures section of Methods (page 13, line 232, ref 45). The details of the source of grading scale have not been added into the results section as well (page 19, line 336). Given the standardized scale which has been used widely for other clinical trials, we do not believe this should be added as a limitation. 

The DAIDS grading table provides an AE severity grading scale ranging from grades 1 to 5 with descriptions for each AE based on the following general guidelines:

 • Grade 1 indicates a mild event

 • Grade 2 indicates a moderate event

 • Grade 3 indicates a severe event 

• Grade 4 indicates a potentially life-threatening event 

• Grade 5 indicates death

3) Completion of sexual history questionnaire as adherence assessments

- Can the authors please clarify how completion of the sexual history questionnaire is used as an adherence assessment method? What is the rationale for this? Has this been validated previously or used in similar studies?

REPLY: Thank you for this comment. The sexual history questionnaire was not used for assessment of adherence, but rather for feasibility (page 12, line 222). Sexual history was collected primarily for the biological correlates of microbiota analysis, which was related to the secondary goal of the trial. Sexual intercourse can significantly influence the composition of the vaginal microbiota. Understanding participants' sexual behaviors allowed us to control for potential confounding factors that could affect the microbiota composition. Tracking sexual behaviors provided a way to ensure that participants adhered to the guidelines, which could otherwise influence the microbiota findings. The sexual history questionnaire was not used directly for assessment of adherence described in the results here.

---

## [Decision Letter · Decision Letter 3]

4 Nov 2024

PONE-D-23-43948R3Feasibility, safety and tolerability of estrogen and/or probiotics for improving vaginal health in Canadian African, Caribbean, and Black women: A pilot phase 1 clinical trialPLOS ONE

Dear Dr. Kaushic,

Thank you for submitting your manuscript to PLOS ONE. After careful consideration, we feel that it has merit but does not fully meet PLOS ONE’s publication criteria as it currently stands. Therefore, we invite you to submit a revised version of the manuscript that addresses the points raised during the review process.

We look forward to receiving your revised manuscript.

Kind regards,

Kazunori Nagasaka

Academic Editor

PLOS ONE

Journal Requirements:

Additional Editor Comments:

Dear Authors,

We apologize for the delay in the peer review process of your paper and thank you for your inquiry regarding the status of your submission.

We acknowledge your research environment and appreciate your patience. However, our reviewers have identified significant concerns regarding the content of your paper,

which prevent us from accepting it at this time. They have highlighted critical statistical issues, particularly concerning the sample size.

We encourage you to discuss these points with the biostatistician on your research team and seek guidance on each of the concerns raised.

We understand that your time is limited, but we would greatly appreciate your attention to these issues and a response addressing the reviewers' comments.

Thank you for your understanding.

Plos One

Kazunori Nagasaka

Reviewers' comments:

Reviewer's Responses to Questions

**Comments to the Author**

1. If the authors have adequately addressed your comments raised in a previous round of review and you feel that this manuscript is now acceptable for publication, you may indicate that here to bypass the “Comments to the Author” section, enter your conflict of interest statement in the “Confidential to Editor” section, and submit your "Accept" recommendation.

Reviewer #3: (No Response)

2. Is the manuscript technically sound, and do the data support the conclusions?

Reviewer #3: No

3. Has the statistical analysis been performed appropriately and rigorously? 

Reviewer #3: No

4. Have the authors made all data underlying the findings in their manuscript fully available?

Reviewer #3: Yes

5. Is the manuscript presented in an intelligible fashion and written in standard English?

Reviewer #3: Yes

6. Review Comments to the Author

Reviewer #3: I think I understand better where you are coming from, but I still believe the statement you have added, "2-5% chance of missing 1:100 AE, based on sample size" is not correct.

In lines 334-366, the numbers reported there are all based on N=29. Assuming an event with a probability of 0.001 or 0.1%, we can calculate the probability of observing exactly zero events in 29 people using the binomial distribution. I'm using R.

> pbinom(0,29,0.001)

[1] 0.9714024

That's a 97% chance of observing exactly zero events, i.e., a 97% chance of missing that event. I'm ok with that because I think it is unreasonable to expect a phase 1 trial to detect such a rare event. Using the same approach, we can see what the probability is of observing exactly zero events when the probability is 0.01:

> pbinom(0,29,0.01)

[1] 0.7471721

That says there is approximately a 75% chance of missing an event, which is far greater than the 2-5% in your statement. If you want to get down to your 2-5%, you need to up the probability by an order of magnitude:

> pbinom(0,29,0.10)

[1] 0.04710129

Thus, you can feel confident about observing an AE in your trial if it occurs in approximately 10% of people.

My contention is that is not good enough. I suspect different fields have different standards, so maybe standards are lower in your field. For instance, I can find a paper that suggests sample sizes have been increasing over time (https://doi.org/10.1093/jnci/dju163) and that your sample size would be on the lower end for studies performed approximately 35 years ago. But maybe those trials are different and there are differences between your field and theirs.

7. PLOS authors have the option to publish the peer review history of their article (what does this mean?). If published, this will include your full peer review and any attached files.

Reviewer #3: No

---

## [Author Response · Author response to Decision Letter 3]

4 Nov 2024

Reply to Reviewers:

Reviewer #3: I think I understand better where you are coming from, but I still believe the statement you have added, "2-5% chance of missing 1:100 AE, based on sample size" is not correct.

In lines 334-366, the numbers reported there are all based on N=29. Assuming an event with a probability of 0.001 or 0.1%, we can calculate the probability of observing exactly zero events in 29 people using the binomial distribution. I'm using R.

> pbinom(0,29,0.001)

[1] 0.9714024

That's a 97% chance of observing exactly zero events, i.e., a 97% chance of missing that event. I'm ok with that because I think it is unreasonable to expect a phase 1 trial to detect such a rare event. Using the same approach, we can see what the probability is of observing exactly zero events when the probability is 0.01:

> pbinom(0,29,0.01)

[1] 0.7471721

That says there is approximately a 75% chance of missing an event, which is far greater than the 2-5% in your statement. If you want to get down to your 2-5%, you need to up the probability by an order of magnitude:

> pbinom(0,29,0.10)

[1] 0.04710129

Thus, you can feel confident about observing an AE in your trial if it occurs in approximately 10% of people.

My contention is that is not good enough. I suspect different fields have different standards, so maybe standards are lower in your field. For instance, I can find a paper that suggests sample sizes have been increasing over time (https://doi.org/10.1093/jnci/dju163) and that your sample size would be on the lower end for studies performed approximately 35 years ago. But maybe those trials are different and there are differences between your field and theirs.

REPLY:

We thank the reviewer for further elaborating on their previous concerns. 

The AE calculations were done on sample size of 37 for each intervention (13+11+13 for estring and 11+13+13 for probiotics, since each intervention was used in 3 out of 4 groups) and not 29. We apologize for the confusion; the number 29 was actually the number of participants who reported AE out of 37. This has been changed and clarified in the manuscript now Line

Although the sample size of 37 is larger than 29, we assume that this will not address the concern the reviewer has raised because change in sample size from 29 to 37 will not change the detection limit of AE by a log fold. Thank you for acknowledging that it is unreasonable to expect a Phase I trial to detect a rare event. We agree that this is pilot Phase I and it was not designed to detect rare adverse events.

Based on reviewer’s comments regarding differences in standards for different fields, we searched some recent Phase I clinical trials in microbiome area to do some comparisons of what is acceptable in this area and found that sample sizes for Phase I clinical trial were typically between 12-33 participants for different trials and spread over multiple study arms. So, our sample size of 43 is very comparable. 

We also found a 2018 FDA document on design of clinical trials as part of drug design process (https://www.fda.gov/patients/drug-development-process/step-3-clinical-research#Clinical_Research_Phase_Studies), which provides typical sample sizes and description for different phases of clinical trials. Below we provide the text from the FDA website, describing Phase I clinical trial.

“Study Participants: 20 to 100 healthy volunteers or people with the disease/condition.

Length of Study: Several months

Purpose: Safety and dosage

During Phase 1 studies, researchers test a new drug in normal volunteers (healthy people). In most cases, 20 to 80 healthy volunteers or people with the disease/condition participate in Phase 1. However, if a new drug is intended for use in cancer patients, researchers conduct Phase 1 studies in patients with that type of cancer.

Phase 1 studies are closely monitored and gather information about how a drug interacts with the human body. Researchers adjust dosing schemes based on animal data to find out how much of a drug the body can tolerate and what its acute side effects are.

As a Phase 1 trial continues, researchers answer research questions related to how it works in the body, the side effects associated with increased dosage, and early information about how effective it is to determine how best to administer the drug to limit risks and maximize possible benefits. This is important to the design of Phase 2 studies.

Approximately 70% of drugs move to the next phase”

Again, our sample size of 43 falls within the range of the description provided above.

We outline below some examples we found of the Phase I clinical trials in the microbiome area 

1. Phase 1 clinical trial evaluating safety, bioavailability, and gut microbiome with a combination of curcumin and ursolic acid in lipid enhanced capsules. Journal of Traditional and Complementary Medicine, 2024. 

Sample size: 18 subjects

https://doi.org/10.1016/j.jtcme.2024.03.002

2. Multiple-Ascending-Dose Phase 1 Clinical Study of the Safety, Tolerability, and Pharmacokinetics of CRS3123, a Narrow-Spectrum Agent with Minimal Disruption of Normal Gut Microbiota Antimicrob Agents Chemother 2019 Dec 20;64(1):e01395-19. 

Sample size: 30 subjects in 3 cohorts of 10 each

doi: 10.1128/AAC.01395-19 

3. A phase 1, randomised, double-blind, placebo-controlled, study to evaluate the safety, tolerability and pharmacokinetics of MAP 315 in healthy adults https://microba.com/news/successful-ibd-phase-i-clinical-trial/ (https://www.anzctr.org.au/Trial/Registration/TrialReview.aspx?id=385380) 

Sample size: 2 cohorts of 16 participants each

4. Randomized Placebo-Controlled, Biomarker-Stratified Phase Ib Microbiome Modulation in Melanoma: Impact of Antibiotic Preconditioning on Microbiome and Immunity 

Cancer Discov (2024) 14 (7): 1161–1175. 

Sample size: 14 patients in 2 arms

https://doi.org/10.1158/2159-8290.CD-24-0066

A couple of specific examples of Phase I trials to test interventions for vaginal microbiome

1.Phase 1 Dose-ranging Safety Trial of Lactobacillus crispatus CTV-05 (LACTIN-V) for the Prevention of Bacterial Vaginosis. Sex Trans Dis. 2010

Sample size: 12 volunteers in 3 blocks of four

https://pmc.ncbi.nlm.nih.gov/articles/PMC2758081/

Below the text from the results of this paper describing Adverse events:

Adverse events

By definition AEs occurred any time after the first applicator was administered and within 35 days of enrollment. All 12 randomized subjects (100%) reported at least one AE. AEs reported in two or more subjects included: vaginal discharge (5 subjects, 42%), abdominal pain (4 subjects, 33%), metrorrhagia (4 subjects, 33%), vulvovaginitis (4 subjects, 33%), vaginal candidiasis (3 subjects, 25%), vaginal odor (3 subjects, 25%) and urinary tract infection (2 subjects, 17%). AEs were mostly mild in severity (41 out of 45 total AEs, 91%) and occurred among 11 subjects. Four of 45 total AEs (9%) were moderate in severity, occurred among 4 subjects and were determined to be unrelated or probably unrelated to study product: in the LACTIN-V arm one woman suffered from gastroenteritis and another woman had a urinary tract infection while in the placebo arm one woman reported ear pain and another woman experienced an upper respiratory tract infection (Table 1). One unexpected AE occurred at the highest dose when an applicator tip separated from the barrel during product administration; this event was not associated with any clinical signs or symptoms. No grade 3 or 4 severity AEs or SAEs occurred during the study.

2.Comparative phase I randomized open-label pilot clinical trial of Gynophilus® (Lcr regenerans®) immediate release capsules versus slow release muco-adhesive tablets

Eur J Clin Microbiol Infect Dis 2018 Jul 21;37(10):1869–1880. 

Sample size: 33 volunteers in 2 groups

doi: 10.1007/s10096-018-3321-8

Below is text from the results section of this paper describing adverse events

Adverse events

No severe adverse events were observed (Table 3). Ten adverse events (AE) were reported, but no differences (p = 0.40) in the frequency of AEs were detected among the 4 arms (Table 3). All AEs were considered not serious. One of the AEs in TRT2 was deemed to be of severe intensity with vulvar itching. Other reported AEs included three that were linked to product use, and the women reported vulvar itching, brown discharges, and vulvovaginal discomfort after administration of the first tablet. However, none of the AEs led to permanent or temporary withdrawal from the study (Table 3).

We hope the above examples will persuade the reviewer that the sample size in our Phase I is very similar and comparable to most other similar Phase I trials widely reported and published in the area of microbiome research. Our Phase I trial was approved by Health Canada and the study design, monitoring and data plan was supervised by CIHR Clinical Trials Network (https://www.hivnet.ubc.ca/).

---

## [Decision Letter · Decision Letter 4]

28 Nov 2024

Feasibility, safety and tolerability of estrogen and/or probiotics for improving vaginal health in Canadian African, Caribbean, and Black women: A pilot phase 1 clinical trial

PONE-D-23-43948R4

Dear Dr. Kaushic,

We’re pleased to inform you that your manuscript has been judged scientifically suitable for publication and will be formally accepted for publication once it meets all outstanding technical requirements.

Kind regards,

Kazunori Nagasaka

Academic Editor

PLOS ONE

Additional Editor Comments (optional):

Dear Authors,

Thank you for submitting your manuscript to Plos One.

We now get two reviewer's comment.

Our reviewers have recommended that the manuscript is ready for publication.

Congratulations on your manuscript and we look forward to receiving your future studies.

Sincerely,

Kazunori Nagasaka

Reviewers' comments:

Reviewer's Responses to Questions

**Comments to the Author**

1. If the authors have adequately addressed your comments raised in a previous round of review and you feel that this manuscript is now acceptable for publication, you may indicate that here to bypass the “Comments to the Author” section, enter your conflict of interest statement in the “Confidential to Editor” section, and submit your "Accept" recommendation.

Reviewer #2: All comments have been addressed

2. Is the manuscript technically sound, and do the data support the conclusions?

Reviewer #2: Yes

3. Has the statistical analysis been performed appropriately and rigorously? 

Reviewer #2: Yes

4. Have the authors made all data underlying the findings in their manuscript fully available?

Reviewer #2: Yes

5. Is the manuscript presented in an intelligible fashion and written in standard English?

Reviewer #2: Yes

6. Review Comments to the Author

Reviewer #2: The authors have addressed all issues, and I believe the manuscript is ready for publication in its current form.

7. PLOS authors have the option to publish the peer review history of their article (what does this mean?). If published, this will include your full peer review and any attached files.

Reviewer #2: No

---

## [Editor Report · Acceptance letter]

4 Dec 2024

PONE-D-23-43948R4 

PLOS ONE

Dear Dr. Kaushic, 

I'm pleased to inform you that your manuscript has been deemed suitable for publication in PLOS ONE. Congratulations! Your manuscript is now being handed over to our production team.

Kind regards, 

on behalf of

Professor Kazunori Nagasaka 

Academic Editor

PLOS ONE